# Relativistic density-functional theory
# based on effective quantum electrodynamics

**Julien Toulouse**[1,2†]

**1** Laboratoire de Chimie Théorique (LCT), Sorbonne Université
and CNRS, F-75005 Paris, France
**2** Institut Universitaire de France, F-75005 Paris, France

† toulouse@lct.jussieu.fr

## Abstract

A relativistic density-functional theory based on a Fock-space effective quantum-electrodynamics (QED) Hamiltonian using the Coulomb or Coulomb-Breit two-particle interaction is developed. This effective QED theory properly includes the effects of vacuum polarization through the creation of electron-positron pairs but does not include explicitly the photon degrees of freedom. It is thus a more tractable alternative to full QED for atomic and molecular calculations. Using the constrained-search formalism, a Kohn-Sham scheme is formulated in a quite similar way to non-relativistic density-functional theory, and some exact properties of the involved density functionals are studied, namely charge-conjugation symmetry and uniform coordinate scaling. The usual no-pair Kohn-Sham scheme is obtained as a well-defined approximation to this relativistic density-functional theory.

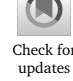

# 1 Introduction

The basic formulation of the relativistic extension of density-functional theory (DFT) was first laid down by generalizing the Hohenberg-Kohn theorem [1] to a Hamiltonian based on quantum electrodynamics (QED) with the internal quantized electromagnetic field and an external classical electromagnetic field [2–5]. These early works did not address the subtle issues of QED renormalization. These issues were studied by Engel, Dreizler, and coworkers [6–10] who put relativistic (current) density-functional theory (RDFT) on more rigorous grounds. In their works, they confirmed the validity of the relativistic extension of the Hohenberg-Kohn theorem using a charge-conjugation-symmetric form of the QED Hamiltonian written with commutators of field operators and appropriate renormalization counterterms. Eschrig *et al.* [11, 12] took another approach to RDFT based on Lieb's Legendre transformation using a normal-ordered QED Hamiltonian. Ohsaku *et al.* [13] proposed a local-density-matrix functional theory based on a QED Hamiltonian with an one-photon-propagator fermion-fermion interaction. Despite these formal foundations of RDFT based on QED, in practice four-component RDFT is invariably applied in the Kohn-Sham (KS) scheme with a non-quantized electromagnetic field and in the no-pair approximation (i.e., neglecting contributions from electron-positron pairs) [14–21], most of the time using non-relativistic exchange-correlation density functionals.

In this work, we examine an alternative RDFT based on a Fock-space effective QED Hamiltonian using the Coulomb or Coulomb-Breit two-particle interaction (see, e.g., Refs. [22–25]). This effective QED theory properly includes the effects of vacuum polarization through the creation of electron-positron pairs but does not include explicitly the photon degrees of freedom. It is thus a more tractable alternative to full QED for atomic and molecular calculations. This so-called no-photon QED has been the subject of a number of detailed mathematical studies [26–31], which in particular established the soundness of this approach at the Hartree-Fock (HF) level. This is thus a good QED level to base a RDFT on. We show that we can develop indeed a RDFT formalism based on this effective QED theory using the constrained-search formalism [32, 33] in a quite similar way to non-relativistic DFT. The usual no-pair KS scheme is then obtained as a well-defined approximation to this RDFT.

The paper is organized as follows. In Section 2, we expose the effective QED theory considered in this work. We define the normal-ordered electron-positron Hamiltonian, we discuss how to define the polarized vacuum state and $N$-negative-charge states by a minimization formulation, and we introduce the no-pair approximation in this approach. In Section 3, we develop a RDFT based on this effective QED theory. We describe the KS scheme in this approach, we give the expression of the Hartree, exchange, and correlation density functionals, we study some exact properties of these functionals, and we discuss the local-density approx-

imation (LDA). Section 4 contains conclusions and perspectives. In the appendices, we prove some important and, to the best of our knowledge, seemingly unknown aspects of the effective QED theory. First, in Appendix A, we show that the electron-positron Hamiltonian expressed in terms of the normal ordering with respect to the free vacuum state has the correct charge-conjugation symmetry. Second, in Appendix B, we show that the electron-positron Hamiltonian based on normal ordering with respect to the free vacuum state is essentially equivalent to an electron-positron Hamiltonian based on commutators and anticommutators of Dirac field operators.

In contrast to the quantum chemistry literature where often everything is formulated in a basis, here we prefer to use a real-space second-quantized formalism which is more adapted to DFT. Hartree atomic units (a.u.) are used throughout the paper.

## 2 Effective quantum electrodynamics

### 2.1 Free Dirac equation and quantized Dirac field

We consider the time-independent free Dirac equation

$$\mathbf{D}(\vec{r})\boldsymbol{\psi}_p(\vec{r}) = \varepsilon_p \boldsymbol{\psi}_p(\vec{r}), \tag{1}$$

with the usual first-quantized $4 \times 4$ Dirac kinetic + rest mass operator

$$\mathbf{D}(\vec{r}) = c\,(\vec{\boldsymbol{\alpha}} \cdot \vec{p}) + \boldsymbol{\beta}\,mc^2, \tag{2}$$

where $\vec{p} = -i\vec{\nabla}$ is the momentum operator, $c = 137.036$ a.u. is the speed of light, $m = 1$ a.u. is the electron mass, and $\vec{\boldsymbol{\alpha}}$ and $\boldsymbol{\beta}$ are the $4 \times 4$ Dirac matrices

$$\vec{\boldsymbol{\alpha}} = \begin{pmatrix} \mathbf{0}_2 & \vec{\boldsymbol{\sigma}} \\ \vec{\boldsymbol{\sigma}} & \mathbf{0}_2 \end{pmatrix} \text{ and } \boldsymbol{\beta} = \begin{pmatrix} \mathbf{I}_2 & \mathbf{0}_2 \\ \mathbf{0}_2 & -\mathbf{I}_2 \end{pmatrix}, \tag{3}$$

where $\vec{\boldsymbol{\sigma}} = (\boldsymbol{\sigma}_x, \boldsymbol{\sigma}_y, \boldsymbol{\sigma}_z)$ is the 3-dimensional vector of the $2 \times 2$ Pauli matrices, and $\mathbf{0}_2$ and $\mathbf{I}_2$ are the $2 \times 2$ zero and identity matrices, respectively.

The eigenfunctions form a set of orthonormal 4-component-spinor orbitals $\{\boldsymbol{\psi}_p\}$ that we will assume as being discretized (by putting the system in a box with periodic boundary conditions). This set can be partitioned into a set of positive-energy orbitals ($\varepsilon_p > 0$) and a set of negative-energy orbitals ($\varepsilon_p < 0$), i.e. $\{\boldsymbol{\psi}_p\} = \{\boldsymbol{\psi}_p\}_{p\in\mathrm{PS}} \cup \{\boldsymbol{\psi}_p\}_{p\in\mathrm{NS}}$, where PS and NS designate the sets of "positive states" and "negative states", respectively. The Dirac field is then quantized as

$$\hat{\boldsymbol{\psi}}(\vec{r}) = \sum_{p\in\mathrm{PS}\cup\mathrm{NS}} \hat{a}_p \boldsymbol{\psi}_p(\vec{r}) = \sum_{p\in\mathrm{PS}} \hat{b}_p \boldsymbol{\psi}_p(\vec{r}) + \sum_{p\in\mathrm{NS}} \hat{d}_p^\dagger \boldsymbol{\psi}_p(\vec{r}), \tag{4}$$

where the sum has been decomposed in a contribution involving electron annihilation operators $\hat{b}_p \equiv \hat{a}_p$ for $p \in \mathrm{PS}$ and a second contribution involving positron creation operators $\hat{d}_p^\dagger \equiv \hat{a}_p$ for $p \in \mathrm{NS}$. The annihilation and creation operators obey the usual fermionic anticommutation rules

$$\{\hat{a}_p, \hat{a}_q^\dagger\} = \delta_{pq} \quad \text{and} \quad \{\hat{a}_p, \hat{a}_q\} = \{\hat{a}_p^\dagger, \hat{a}_q^\dagger\} = 0 \quad \text{for } p, q \in \mathrm{PS} \cup \mathrm{NS}, \tag{5}$$

and the corresponding free vacuum state $|0\rangle$ is defined such that

$$\hat{b}_p|0\rangle = 0 \quad \text{for } p \in \mathrm{PS} \quad \text{and} \quad \hat{d}_p|0\rangle = 0 \quad \text{for } p \in \mathrm{NS}. \tag{6}$$

## 2.2 Electron-positron Hamiltonian

We then consider the normal-ordered electron-positron Hamiltonian in Fock space written with this quantized Dirac field introduced in Refs. [22, 34] (see, also, Ref. [23]) that we can write as

$$\hat{H} = \hat{T}_{\mathrm{D}} + \hat{W} + \hat{V}, \tag{7}$$

where the Dirac kinetic + rest mass operator $\hat{T}_{\mathrm{D}}$, the two-particle interaction operator $\hat{W}$, and the external potential-energy interaction operator $\hat{V}$ are expressed as (using $\sigma$, $\rho$, $\tau$, $\upsilon$ as spinor indices ranging from 1 to 4)

$$\hat{T}_{\mathrm{D}} = \int \mathrm{Tr}[\mathbf{D}(\vec{r})\hat{\mathbf{n}}_1(\vec{r},\vec{r}\,')]_{\vec{r}\,'=\vec{r}} \, \mathrm{d}\vec{r} \equiv \sum_{\sigma\rho} \int [D_{\sigma\rho}(\vec{r})\hat{n}_{1,\rho\sigma}(\vec{r},\vec{r}\,')]_{\vec{r}\,'=\vec{r}} \, \mathrm{d}\vec{r}, \tag{8}$$

and

$$
\begin{aligned}
\hat{W} &= \frac{1}{2} \iint \mathrm{Tr}[\mathbf{w}(\vec{r}_1,\vec{r}_2)\hat{\mathbf{n}}_2(\vec{r}_1,\vec{r}_2)]\mathrm{d}\vec{r}_1\mathrm{d}\vec{r}_2 \\
&\equiv \frac{1}{2} \sum_{\sigma\rho\tau\upsilon} \iint w_{\sigma\tau\rho\upsilon}(\vec{r}_1,\vec{r}_2)\hat{n}_{2,\rho\upsilon\sigma\tau}(\vec{r}_1,\vec{r}_2) \, \mathrm{d}\vec{r}_1\mathrm{d}\vec{r}_2,
\end{aligned}
\tag{9}
$$

and

$$\hat{V} = \int v(\vec{r})\hat{n}(\vec{r}) \, \mathrm{d}\vec{r}, \tag{10}$$

where the one-particle density-matrix operator $\hat{\mathbf{n}}_1(\vec{r},\vec{r}\,')$ and the pair density-matrix operator $\hat{\mathbf{n}}_2(\vec{r}_1,\vec{r}_2)$ are defined using creation and annihilation Dirac field operators with normal ordering $\mathcal{N}[...]$ of the elementary creation and annihilation operators $\hat{b}_p^\dagger$, $\hat{b}_p$, $\hat{d}_p^\dagger$, $\hat{d}_p$ with respect to the free vacuum state $|0\rangle$

$$\hat{n}_{1,\rho\sigma}(\vec{r},\vec{r}\,') = \mathcal{N}[\hat{\psi}_\sigma^\dagger(\vec{r}\,')\hat{\psi}_\rho(\vec{r})], \tag{11}$$

$$\hat{n}_{2,\rho\upsilon\sigma\tau}(\vec{r}_1,\vec{r}_2) = \mathcal{N}[\hat{\psi}_\tau^\dagger(\vec{r}_2)\hat{\psi}_\sigma^\dagger(\vec{r}_1)\hat{\psi}_\rho(\vec{r}_1)\hat{\psi}_\upsilon(\vec{r}_2)], \tag{12}$$

and the opposite charge density operator is

$$\hat{n}(\vec{r}) = \mathrm{Tr}[\hat{\mathbf{n}}(\vec{r})] \equiv \sum_\sigma \hat{n}_{\sigma\sigma}(\vec{r}), \tag{13}$$

where $\hat{\mathbf{n}}(\vec{r}) = \hat{\mathbf{n}}_1(\vec{r},\vec{r})$. Note that, in the non-relativistic theory, the opposite charge density operator reduces to the usual one-electron density operator, which is why we prefer to use the opposite charge density operator $\hat{n}(\vec{r})$ and not the charge density operator $\hat{\rho}(\vec{r}) = -\hat{n}(\vec{r})$. The normal ordering in the definition of the Dirac kinetic + rest mass operator $\hat{T}_{\mathrm{D}}$ in Eq. (8) ensures that this operator is bounded from below with a nonnegative spectrum. In Eq. (9) $\mathbf{w}(\vec{r}_1,\vec{r}_2)$ is a two-particle interaction matrix potential which could be for example the Coulomb (C) + Breit (B) interaction

$$w_{\sigma\tau\rho\upsilon}(\vec{r}_1,\vec{r}_2) = w_{\sigma\tau\rho\upsilon}^{\mathrm{C}}(r_{12}) + w_{\sigma\tau\rho\upsilon}^{\mathrm{B}}(\vec{r}_{12}), \tag{14}$$

with $\vec{r}_{12} = \vec{r}_1 - \vec{r}_2$ and $r_{12} = |\vec{r}_{12}|$, and

$$w_{\sigma\tau\rho\upsilon}^{\mathrm{C}}(r_{12}) = w(r_{12})\delta_{\sigma\rho}\delta_{\tau\upsilon}, \tag{15}$$

$$w^{\mathrm{B}}_{\sigma\tau\rho\upsilon}(\vec{r}_{12}) = -\frac{1}{2}w(r_{12})\left(\vec{\alpha}_{\sigma\rho}\cdot\vec{\alpha}_{\tau\upsilon} + \frac{(\vec{\alpha}_{\sigma\rho}\cdot\vec{r}_{12})(\vec{\alpha}_{\tau\upsilon}\cdot\vec{r}_{12})}{r_{12}^2}\right), \tag{16}$$

where $w(r_{12}) = 1/r_{12}$. The Coulomb-Breit interaction corresponds to the single-photon exchange electron-electron scattering amplitude in QED evaluated with the zero-frequency limit of the photon propagator in the Coulomb electromagnetic gauge. More specifically, the instantaneous Coulomb interaction corresponds to the longitudinal component of the photon propagator, whereas the Breit interaction is obtained from the zero-frequency transverse component of the photon propagator. The Breit interaction comprises the instantaneous magnetic Gaunt interaction, $-w(r_{12})\vec{\alpha}_{\sigma\rho}\cdot\vec{\alpha}_{\tau\upsilon}$, and the remaining lowest-order retardation correction (see, e.g., Ref. [35]). In Eq. (10) $v(\vec{r})$ is an external scalar potential, e.g. the Coulomb potential generated by the nuclei. For simplicity and following the most common framework used for molecular calculations, we do not consider the case of an external vector potential. Due to the external potential [Eq. (10)] and Coulomb-Breit two-particle interaction [Eq. (9)], the present theory is not Lorentz invariant, which is in the spirit in which relativistic molecular calculations are carried out presently.

The electron-positron Hamiltonian $\hat{H}$ does not commute separately with the electron and positron number operators,

$$\hat{N}_{\mathrm{e}} = \sum_{p\in\mathrm{PS}}\hat{b}_p^\dagger\hat{b}_p \quad\text{and}\quad \hat{N}_{\mathrm{p}} = \sum_{p\in\mathrm{NS}}\hat{d}_p^\dagger\hat{d}_p\,, \tag{17}$$

i.e., it does not conserve electron or positron numbers. However, the Hamiltonian $\hat{H}$ commutes with the opposite charge operator (or electron-excess number operator)

$$\hat{N} = \hat{N}_{\mathrm{e}} - \hat{N}_{\mathrm{p}}\,, \tag{18}$$

i.e., it conserves charge. As a consequence, the eigenstates of the Hamiltonian $\hat{H}$ belongs to the Fock space gathering together different particle-number sectors

$$\mathcal{F} = \bigoplus_{(N_{\mathrm{e}},N_{\mathrm{p}})=(0,0)}^{(\infty,\infty)} \mathcal{H}^{(N_{\mathrm{e}},N_{\mathrm{p}})}\,, \tag{19}$$

where $\mathcal{H}^{(N_{\mathrm{e}},N_{\mathrm{p}})}$ is the Hilbert space for $N_{\mathrm{e}}$ electrons and $N_{\mathrm{p}}$ positrons, and $\oplus$ designates the direct sum. The Fock space can also be decomposed into charge sectors

$$\mathcal{F} = \bigoplus_{q=-\infty}^{\infty} \mathcal{H}_q\,, \tag{20}$$

where $\mathcal{H}_q$ is the Hilbert space for opposite charge $q$. For $q \geq 0$, we have $\mathcal{H}_q = \mathcal{H}^{(q,0)} \oplus \mathcal{H}^{(q+1,1)} \oplus \mathcal{H}^{(q+2,2)} \oplus \cdots \oplus \mathcal{H}^{(q+\infty,\infty)}$, and for $q \leq 0$, we have $\mathcal{H}_q = \mathcal{H}^{(0,|q|)} \oplus \mathcal{H}^{(1,|q|+1)} \oplus \mathcal{H}^{(2,|q|+2)} \oplus \cdots \oplus \mathcal{H}^{(\infty,|q|+\infty)}$.

Importantly, due to the fact that the electron-positron Hamiltonian in Eq. (7) is expressed with normal ordering with respect to the free vacuum state, it has the correct charge-conjugation symmetry, i.e. $\hat{C}\hat{H}[v]\hat{C}^\dagger = \hat{H}[-v]$ where $\hat{H}[v]$ is the Hamiltonian in Eq. (7) with an arbitrary external potential $v$ and $\hat{C}$ is the charge-conjugation operator in Fock space (see Appendix A).

## 2.3 No-particle vacuum states

By construction of the Hamiltonian $\hat{H}$, the free vacuum state $|0\rangle$ has a zero energy, i.e. $E_0^{\mathrm{free}} = \langle 0|\hat{H}|0\rangle = 0$. However, this is generally not the lowest-energy vacuum state. We

can consider other no-particle vacuum states $|\tilde{0}\rangle$ (often referred to as polarized vacuum or dressed vacuum) parametrized as [23, 36] (see, also, Refs. [22, 34, 37, 38])

$$|\tilde{0}\rangle = e^{\hat{\kappa}}|0\rangle\,, \tag{21}$$

where $e^{\hat{\kappa}}$ performs an orbital rotation in Fock space (corresponding to a Bogoliubov transformation mixing electron annihilation operators $\hat{b}_p$ and positron creation operators $\hat{d}_p^\dagger$ [22]) with the anti-Hermitian operator $\hat{\kappa}$

$$\hat{\kappa} = \sum_{p,q\in\text{PS}\cup\text{NS}} \kappa_{pq}\hat{a}_p^\dagger\hat{a}_q \;\; = \;\; \sum_{p,q\in\text{PS}} \kappa_{pq}\hat{b}_p^\dagger\hat{b}_q + \sum_{p\in\text{PS}}\sum_{q\in\text{NS}} \kappa_{pq}\hat{b}_p^\dagger\hat{d}_q^\dagger$$
$$+ \sum_{p\in\text{NS}}\sum_{q\in\text{PS}} \kappa_{pq}\hat{d}_p\hat{b}_q + \sum_{p,q\in\text{NS}} \kappa_{pq}\hat{d}_p\hat{d}_q^\dagger\,, \tag{22}$$

with the orbital rotation parameters $\kappa_{pq} \in \mathbb{C}$ being the elements of an anti-Hermitian matrix $\boldsymbol{\kappa}$. Note that the second term in the last expression of Eq. (22) creates electron-positron pairs. This generates new creation and annihilation operators related to the original ones via the unitary matrix $\mathbf{U} = e^{\boldsymbol{\kappa}}$

$$\hat{\tilde{a}}_p^\dagger = e^{\hat{\kappa}}\hat{a}_p^\dagger e^{-\hat{\kappa}} = \sum_{q\in\text{PS}\cup\text{NS}} \hat{a}_q^\dagger U_{qp} \quad \text{and} \quad \hat{\tilde{a}}_p = e^{\hat{\kappa}}\hat{a}_p e^{-\hat{\kappa}} = \sum_{q\in\text{PS}\cup\text{NS}} \hat{a}_q U_{qp}^* \quad \text{for } p \in \text{PS}\cup\text{NS}, \tag{23}$$

and corresponding new orbitals

$$\tilde{\psi}_p(\vec{r}) = \sum_{q\in\text{PS}\cup\text{NS}} \psi_q(\vec{r})U_{qp} \quad \text{for } p \in \text{PS}\cup\text{NS}, \tag{24}$$

such that the Dirac field operator in Eq. (4) can be rewritten as

$$\hat{\psi}(\vec{r}) \;\; = \;\; \sum_{p\in\text{PS}\cup\text{NS}} \hat{\tilde{a}}_p\tilde{\psi}_p(\vec{r}) = \sum_{p\in\text{PS}} \hat{\tilde{b}}_p\tilde{\psi}_p(\vec{r}) + \sum_{p\in\text{NS}} \hat{\tilde{d}}_p^\dagger\tilde{\psi}_p(\vec{r})\,, \tag{25}$$

with again $\hat{\tilde{b}}_p \equiv \hat{\tilde{a}}_p$ for $p \in \text{PS}$ and $\hat{\tilde{d}}_p^\dagger \equiv \hat{\tilde{a}}_p$ for $p \in \text{NS}$. The new creation and annihilation operators still obey the fermionic anticommutation rules in Eq. (5). Moreover, even though this orbital rotation does not necessarily preserve the sign of the orbital energies, it does preserve the charge, i.e. we have $[\hat{N}, \hat{\tilde{b}}_p^\dagger] = \hat{\tilde{b}}_p^\dagger$ and $[\hat{N}, \hat{\tilde{d}}_p^\dagger] = -\hat{\tilde{d}}_p^\dagger$. So the new creation operators $\hat{\tilde{b}}_p^\dagger$ and $\hat{\tilde{d}}_p^\dagger$ can still be interpreted as creating electrons and positrons, respectively, and the partition into PS and NS sets should now be understood as a partition into positive and negative opposite charge states. As expected, the new electron and positron annihilation operators satisfy

$$\hat{\tilde{b}}_p|\tilde{0}\rangle = 0 \;\; \text{for } p \in \text{PS} \quad \text{and} \quad \hat{\tilde{d}}_p|\tilde{0}\rangle = 0 \;\; \text{for } p \in \text{NS}. \tag{26}$$

The new vacuum state $|\tilde{0}\rangle$ contains electron-positron pairs associated with the original operators $\hat{b}_p^\dagger$ and $\hat{d}_p^\dagger$ but does not contain any particle associated with the new operators $\hat{\tilde{b}}_p^\dagger$ and $\hat{\tilde{d}}_p^\dagger$.

We can then introduce a new one-particle density-matrix operator $\hat{\tilde{\mathbf{n}}}_1(\vec{r}, \vec{r}\,')$ and a new pair density-matrix operator $\hat{\tilde{\mathbf{n}}}_2(\vec{r}_1, \vec{r}_2)$ defined using normal ordering $\tilde{\mathcal{N}}[...]$ of the new elementary creation and annihilation operators $\hat{\tilde{b}}_p^\dagger$, $\hat{\tilde{b}}_p$, $\hat{\tilde{d}}_p^\dagger$, $\hat{\tilde{d}}_p$ with respect to the new vacuum state $|\tilde{0}\rangle$

$$\hat{\tilde{n}}_{1,\rho\sigma}(\vec{r}, \vec{r}\,') = \tilde{\mathcal{N}}[\hat{\psi}_\sigma^\dagger(\vec{r}\,')\hat{\psi}_\rho(\vec{r})]\,, \tag{27}$$

and

$$\hat{\tilde{n}}_{2,\rho\upsilon\sigma\tau}(\vec{r}_1, \vec{r}_2) = \tilde{\mathcal{N}}[\hat{\psi}_\tau^\dagger(\vec{r}_2)\hat{\psi}_\sigma^\dagger(\vec{r}_1)\hat{\psi}_\rho(\vec{r}_1)\hat{\psi}_\upsilon(\vec{r}_2)]\,. \tag{28}$$

Using Wick's theorem, the original one-particle density-matrix and pair density-matrix operators in Eq. (11) and (12) can be rewritten as [22]

$$\hat{n}_{1,\rho\sigma}(\vec{r},\vec{r}\,') = \hat{\tilde{n}}_{1,\rho\sigma}(\vec{r},\vec{r}\,') + \tilde{n}_{1,\rho\sigma}^{\mathrm{vp}}(\vec{r},\vec{r}\,'), \tag{29}$$

and

$$\begin{aligned}
\hat{n}_{2,\rho\upsilon\sigma\tau}(\vec{r}_1,\vec{r}_2) &= \hat{\tilde{n}}_{2,\rho\upsilon\sigma\tau}(\vec{r}_1,\vec{r}_2) + \tilde{n}_{1,\upsilon\tau}^{\mathrm{vp}}(\vec{r}_2,\vec{r}_2)\hat{\tilde{n}}_{1,\rho\sigma}(\vec{r}_1,\vec{r}_1) + \tilde{n}_{1,\rho\sigma}^{\mathrm{vp}}(\vec{r}_1,\vec{r}_1)\hat{\tilde{n}}_{1,\upsilon\tau}(\vec{r}_2,\vec{r}_2) \\
&\quad - \tilde{n}_{1,\upsilon\sigma}^{\mathrm{vp}}(\vec{r}_2,\vec{r}_1)\hat{\tilde{n}}_{1,\rho\tau}(\vec{r}_1,\vec{r}_2) - \tilde{n}_{1,\rho\tau}^{\mathrm{vp}}(\vec{r}_1,\vec{r}_2)\hat{\tilde{n}}_{1,\upsilon\sigma}(\vec{r}_2,\vec{r}_1) + \tilde{n}_{2,\rho\upsilon\sigma\tau}^{\mathrm{vp}}(\vec{r}_1,\vec{r}_2),
\end{aligned} \tag{30}$$

where $\tilde{\mathbf{n}}_1^{\mathrm{vp}}(\vec{r},\vec{r}\,')$ is the vacuum-polarization (vp) one-particle density matrix

$$\begin{aligned}
\tilde{n}_{1,\rho\sigma}^{\mathrm{vp}}(\vec{r},\vec{r}\,') &= \langle\tilde{0}|\hat{n}_{1,\rho\sigma}(\vec{r},\vec{r}\,')|\tilde{0}\rangle \\
&= \langle\tilde{0}|\hat{\psi}_\sigma^\dagger(\vec{r}\,')\hat{\psi}_\rho(\vec{r})|\tilde{0}\rangle - \langle 0|\hat{\psi}_\sigma^\dagger(\vec{r}\,')\hat{\psi}_\rho(\vec{r})|0\rangle \\
&= \sum_{p\in\mathrm{NS}}\tilde{\psi}_{p,\sigma}^*(\vec{r}\,')\tilde{\psi}_{p,\rho}(\vec{r}) - \sum_{p\in\mathrm{NS}}\psi_{p,\sigma}^*(\vec{r}\,')\psi_{p,\rho}(\vec{r}),
\end{aligned} \tag{31}$$

and $\tilde{\mathbf{n}}_2^{\mathrm{vp}}(\vec{r}_1,\vec{r}_2)$ is the vacuum-polarization pair-density matrix

$$\tilde{n}_{2,\rho\upsilon\sigma\tau}^{\mathrm{vp}}(\vec{r}_1,\vec{r}_2) = \tilde{n}_{1,\upsilon\tau}^{\mathrm{vp}}(\vec{r}_2,\vec{r}_2)\tilde{n}_{1,\rho\sigma}^{\mathrm{vp}}(\vec{r}_1,\vec{r}_1) - \tilde{n}_{1,\rho\tau}^{\mathrm{vp}}(\vec{r}_1,\vec{r}_2)\tilde{n}_{1,\upsilon\sigma}^{\mathrm{vp}}(\vec{r}_2,\vec{r}_1). \tag{32}$$

The electron-positron Hamiltonian in Eq. (7) can then be rewritten as [22]

$$\hat{H} = \hat{\tilde{T}}_{\mathrm{D}} + \hat{\tilde{W}} + \hat{\tilde{V}} + \hat{\tilde{V}}^{\mathrm{vp}} + \tilde{E}_0, \tag{33}$$

with

$$\hat{\tilde{T}}_{\mathrm{D}} = \int \mathrm{Tr}[\mathbf{D}(\vec{r})\hat{\tilde{\mathbf{n}}}_1(\vec{r},\vec{r}\,')]_{\vec{r}\,'=\vec{r}}\,\mathrm{d}\vec{r}, \tag{34}$$

and

$$\hat{\tilde{W}} = \frac{1}{2}\iint \mathrm{Tr}[\mathbf{w}(\vec{r}_1,\vec{r}_2)\hat{\tilde{\mathbf{n}}}_2(\vec{r}_1,\vec{r}_2)]\mathrm{d}\vec{r}_1\mathrm{d}\vec{r}_2, \tag{35}$$

and

$$\hat{\tilde{V}} = \int \nu(\vec{r})\hat{\tilde{n}}(\vec{r})\,\mathrm{d}\vec{r}, \tag{36}$$

with the new opposite charge density operator

$$\hat{\tilde{n}}(\vec{r}) = \mathrm{Tr}[\hat{\tilde{\mathbf{n}}}(\vec{r})], \tag{37}$$

where $\hat{\tilde{\mathbf{n}}}(\vec{r}) = \hat{\tilde{\mathbf{n}}}_1(\vec{r},\vec{r})$. In Eq. (33), the normal reordering with respect to the new vacuum state $|\tilde{0}\rangle$ [Eqs. (29) and (30)] has generated two new terms: the vacuum-polarization potential operator $\hat{\tilde{V}}^{\mathrm{vp}}$ and the new vacuum energy $\tilde{E}_0$. The vacuum-polarization potential operator [22] can be written as

$$\hat{\tilde{V}}^{\mathrm{vp}} = \hat{\tilde{V}}_{\mathrm{H}}^{\mathrm{vp}} + \hat{\tilde{V}}_{\mathrm{x}}^{\mathrm{vp}}, \tag{38}$$

with a Hartree (or direct) contribution

$$\hat{\tilde{V}}_{\mathrm{H}}^{\mathrm{vp}} = \int \mathrm{Tr}[\tilde{\mathbf{v}}_{\mathrm{H}}^{\mathrm{vp}}(\vec{r})\hat{\tilde{\mathbf{n}}}(\vec{r})]\mathrm{d}\vec{r} \equiv \sum_{\rho\sigma}\int \tilde{v}_{\mathrm{H},\sigma\rho}^{\mathrm{vp}}(\vec{r})\hat{\tilde{n}}_{\rho\sigma}(\vec{r})\mathrm{d}\vec{r}, \tag{39}$$

where $\tilde{v}_{\mathrm{H},\sigma\rho}^{\mathrm{vp}}(\vec{r}_1) = \sum_{\tau\upsilon}\int w_{\sigma\tau\rho\upsilon}(\vec{r}_1,\vec{r}_2)\tilde{n}_{\upsilon\tau}^{\mathrm{vp}}(\vec{r}_2)\mathrm{d}\vec{r}_2$ and $\tilde{n}_{\upsilon\tau}^{\mathrm{vp}}(\vec{r}_2) = \tilde{n}_{1,\upsilon\tau}^{\mathrm{vp}}(\vec{r}_2,\vec{r}_2)$, and an exchange contribution

$$\hat{\tilde{V}}_{\mathrm{x}}^{\mathrm{vp}} = \iint \mathrm{Tr}[\tilde{\mathbf{v}}_{\mathrm{x}}^{\mathrm{vp}}(\vec{r}_1,\vec{r}_2)\hat{\tilde{\mathbf{n}}}_1(\vec{r}_1,\vec{r}_2)]\mathrm{d}\vec{r}_1\mathrm{d}\vec{r}_2\,, \tag{40}$$

where $\tilde{v}_{\mathrm{x},\tau\rho}^{\mathrm{vp}}(\vec{r}_1,\vec{r}_2) = -\sum_{\sigma\upsilon}w_{\sigma\tau\rho\upsilon}(\vec{r}_1,\vec{r}_2)\tilde{n}_{1,\upsilon\sigma}^{\mathrm{vp}}(\vec{r}_2,\vec{r}_1)$. Note that in the literature the name "vacuum polarization" is often restricted to the direct term in Eq. (39) whereas the exchange term in Eq. (40) is often designated as "self-energy" (see, e.g., Ref. [25]). Here, we adopt the terminology of Ref. [22] where vacuum polarization designates both terms. Finally, the new no-particle vacuum energy [22] can be written as

$$\begin{aligned}\tilde{E}_0 = \langle\tilde{0}|\hat{H}|\tilde{0}\rangle &= \int \mathrm{Tr}[\mathbf{D}(\vec{r})\tilde{\mathbf{n}}_1^{\mathrm{vp}}(\vec{r},\vec{r}\,')]_{\vec{r}\,'=\vec{r}}\,\mathrm{d}\vec{r} + \int \nu(\vec{r})\tilde{n}^{\mathrm{vp}}(\vec{r})\,\mathrm{d}\vec{r} \\ &\quad + \frac{1}{2}\iint \mathrm{Tr}[\mathbf{w}(\vec{r}_1,\vec{r}_2)\tilde{\mathbf{n}}_2^{\mathrm{vp}}(\vec{r}_1,\vec{r}_2)]\mathrm{d}\vec{r}_1\mathrm{d}\vec{r}_2\,. \end{aligned} \tag{41}$$

Throughout the paper, $|\tilde{0}\rangle$ will refer to an arbitrary vacuum state, often referred to as floating vacuum, and $\{\tilde{\psi}_p\}$ and $\tilde{E}_0$ will refer to its associated orbitals and vacuum energy. The optimal HF vacuum state is defined as the vacuum state minimizing $\tilde{E}_0$ with respect to the orbital rotation parameters $\boldsymbol{\kappa}$

$$E_0^{\mathrm{HF}} = \min_{\boldsymbol{\kappa}} \tilde{E}_0\,. \tag{42}$$

Clearly, if $\mathbf{n}_1^{\mathrm{vp}}(\vec{r},\vec{r}\,') = \mathbf{0}$ then $\tilde{E}_0 = 0$, and thus $E_0^{\mathrm{HF}}$ is necessarily negative. It can in fact diverges to $-\infty$ due to infrared (IR) and ultraviolet (UV) divergences. The IR divergences appear when taking the continuum limit of the sums in Eq. (31), but can simply be avoided by putting the system in a box with periodic boundary conditions and taking the thermodynamic limit of quantities per volume unit (see, e.g., Refs. [11, 29, 30]), similarly to what is done for the homogeneous electron gas. The UV divergences come from the unbound large-energy (or large index $p$) limit of each sum in Eq. (31), even if we expect a cancellation of these UV divergences to some extent between the two sums. A standard way of dealing with these UV divergences is to introduce a fixed UV momentum cutoff and to remove the cutoff dependence via renormalization of the electron charge and mass in the Hamiltonian [26–31, 39] (see also Ref. [40]). We leave for future work these subtle issues and simply assume in the rest of this work that a proper renormalization scheme is applied in order to keep everything finite.

Finally, in Appendix B, we provide an alternative definition of the electron-positron Hamiltonian based on commutators and anticommutators of Dirac field operators and we show that, after removing the vacuum energy, both Hamiltonians are equivalent to each other and also identical to the effective QED Hamiltonian of Refs. [25, 41–45] [see Eq. (176)].

## 2.4 Correlated vacuum state

More generally, the vacuum state can be defined beyond the HF approximation as the lowest-energy state with zero charge, which will refer to as the correlated vacuum state $|\Psi_0\rangle \in \mathcal{H}_0$. In a full configuration-interaction approach, the correlated vacuum state can be parametrized as a linear combination of states with arbitrary numbers of electron-positron pairs

$$\begin{aligned}|\Psi_0\rangle &= \Bigg(c_0 + \sum_{p_1\in\mathrm{PS}}\sum_{q_1\in\mathrm{NS}} c_{p_1 q_1}\hat{b}_{p_1}^\dagger\hat{d}_{q_1}^\dagger + \sum_{p_1,p_2\in\mathrm{PS}}\sum_{q_1,q_2\in\mathrm{NS}} c_{p_1 q_1 p_2 q_2}\hat{b}_{p_1}^\dagger\hat{d}_{q_1}^\dagger\hat{b}_{p_2}^\dagger\hat{d}_{q_2}^\dagger \\ &\quad + \sum_{p_1,p_2,p_3\in\mathrm{PS}}\sum_{q_1,q_2,q_3\in\mathrm{NS}} c_{p_1 q_1 p_2 q_2 p_3 q_3}\hat{b}_{p_1}^\dagger\hat{d}_{q_1}^\dagger\hat{b}_{p_2}^\dagger\hat{d}_{q_2}^\dagger\hat{b}_{p_3}^\dagger\hat{d}_{q_3}^\dagger + \cdots\Bigg)|0\rangle\,, \end{aligned} \tag{43}$$

and minimizing the energy with respect to the coefficients leads to the correlated vacuum energy $E_0 = \langle \Psi_0 | \hat{H} | \Psi_0 \rangle$. Note that the particles inside this vacuum state cannot generally be absorbed into an orbital rotation because of the two-particle interaction in the Hamiltonian. Therefore, the correlated vacuum state generally contains electron-positron pairs, in the same way as the non-relativistic ground state contains excited Slater determinants that cannot be absorbed into a redefinition of the orbitals. With the parametrization of the vacuum state in Eq. (43), there is no need to perform orbital rotations (i.e., orbital rotation parameters are redundant). The correlated vacuum state $|\Psi_0\rangle$ and correlated vacuum energy $E_0$ include all vacuum contributions (i.e., contributions from orbitals in the set NS) to all orders in the two-particle interaction.

## 2.5  $N$-negative-charge states

The ground-state energy for a net total amount of $q = N$ negative charges (the equivalent of $N$ electrons for the non-relativistic theory) is found as

$$E_N = \min_{|\Psi\rangle \in \mathcal{H}_N} \langle \Psi | \hat{T}_D + \hat{W} + \hat{V} | \Psi \rangle, \tag{44}$$

where $|\Psi\rangle$ is constrained to have a net total amount of $N$ negative charges, i.e. $\int \langle \Psi | \hat{n}(\vec{r}) | \Psi \rangle \mathrm{d}\vec{r} = N$. Note that we will always tacitly assume that $|\Psi\rangle$ is constrained to be normalized to 1, i.e. $\langle \Psi | \Psi \rangle = 1$. A state $|\Psi\rangle \in \mathcal{H}_N$ has the form

$$|\Psi\rangle = \left( \sum_{p_1,\ldots,p_N \in \mathrm{PS}} c_{p_1 \ldots p_N} \hat{b}_{p_1}^\dagger \cdots \hat{b}_{p_N}^\dagger + \sum_{p_1,\ldots,p_N,p_{N+1} \in \mathrm{PS}} \sum_{q_1 \in \mathrm{NS}} c_{p_1 \ldots p_N p_{N+1} q_1} \hat{b}_{p_1}^\dagger \cdots \hat{b}_{p_N}^\dagger \hat{b}_{p_{N+1}}^\dagger \hat{d}_{q_1}^\dagger \right.$$
$$\left. + \sum_{p_1,\ldots,p_N,p_{N+1},p_{N+2} \in \mathrm{PS}} \sum_{q_1,q_2 \in \mathrm{NS}} c_{p_1 \ldots p_N p_{N+1} q_1 p_{N+2} q_2} \hat{b}_{p_1}^\dagger \cdots \hat{b}_{p_N}^\dagger \hat{b}_{p_{N+1}}^\dagger \hat{d}_{q_1}^\dagger \hat{b}_{p_{N+2}}^\dagger \hat{d}_{q_2}^\dagger + \cdots \right) |0\rangle. \tag{45}$$

Again, vacuum contributions to all orders are included in the presence of $N$ negative charges, and there is no need to perform orbital rotations. Obviously, in the special case $N = 0$, this reduces to the correlated vacuum state in Eq. (43).

Since the number of particles is not fixed for the Fock state $|\Psi\rangle$ in Eq. (45), there is no concept of $N$-particle wave function (depending on $N$ space coordinates) associated with the state $|\Psi\rangle$. Thus, one cannot study for example the wave function at electron-electron coalescence. However, one could study the small interparticle behavior of the pair-density matrix $\mathbf{n}_2(\vec{r}_1, \vec{r}_2) = \langle \Psi | \hat{n}_2(\vec{r}_1, \vec{r}_2) | \Psi \rangle$, which should ultimately control the convergence rate of the energy with respect to the one-particle basis used to expand the orbitals. So far, as far as we know, the electron-electron coalescence has been studied only for more approximate configuration-space-based relativistic theories where the concept of wave function is retained [46,47]. How to extend in practice these studies to Fock-space approaches such as the one of the present work is an open question.

Finally, let us mention that we can allow for negative $N$ to describe the case of $N$-positive-charge states, i.e. states with a majority of positrons. We will however normally think of $N$ as being positive and write the equations accordingly.

## 2.6  No-pair approximation

Finally, we consider the no-pair (np) approximation [48,49]. In the context of the present theory, it is natural to first define what we will call here a "no-pair with vacuum-polarization"

(npvp) approximation (see Ref. [22]) in which the ground-state energy for $N$ electrons is expressed as

$$E_N^{\text{npvp}} = \min_{|\Psi_+\rangle \in \tilde{\mathcal{H}}^{(N,0)}} \langle \Psi_+ | \hat{T}_{\text{D}} + \hat{W} + \hat{V} | \Psi_+ \rangle, \tag{46}$$

where the minimization is over normalized states in the set that we designate by $\tilde{\mathcal{H}}^{(N,0)} \equiv e^{\hat{\kappa}} \mathcal{H}^{(N,0)}$ which is the set of states generated by all orbital rotations of $N$-electron states. A state $|\Psi_+\rangle \in \tilde{\mathcal{H}}^{(N,0)}$ has the form

$$|\Psi_+\rangle = e^{\hat{\kappa}} \sum_{p_1,\dots,p_N \in \text{PS}} c_{p_1\dots p_N} \hat{b}_{p_1}^\dagger \cdots \hat{b}_{p_N}^\dagger |0\rangle = \sum_{p_1,\dots,p_N \in \text{PS}} c_{p_1\dots p_N} \tilde{\hat{b}}_{p_1}^\dagger \cdots \tilde{\hat{b}}_{p_N}^\dagger |\tilde{0}\rangle. \tag{47}$$

We can also write this state as

$$|\Psi_+\rangle = \tilde{\hat{P}}_+ |\Psi\rangle, \tag{48}$$

where $|\Psi\rangle$ is an arbitrary state constrained to have a net total amount of $N$ negative charges, i.e. $|\Psi\rangle \in \mathcal{H}_N$, and $\tilde{\hat{P}}_+$ is the projector onto the $N$-electron Hilbert space constructed from the set of electron creation operators $\{\tilde{\hat{b}}_p^\dagger\}$ associated with a floating vacuum state $|\tilde{0}\rangle$. The energy is not only minimized with respect to $|\Psi\rangle$ but also with respect to the projector $\tilde{\hat{P}}_+$ by performing orbital rotations between PS and NS orbitals. The optimal floating vacuum state $|\tilde{0}\rangle$ will of course depend on the number of electrons $N$ considered. This npvp approximation thus restores the concept of the $N$-electron ($4N$-component spinor) wave function, i.e.

$$\Psi_+(\vec{r}_1, \vec{r}_2, \dots, \vec{r}_N) = \sum_{p_1,\dots,p_N \in \text{PS}} c_{p_1\dots p_N} \tilde{\psi}_{p_1}(\vec{r}_1) \wedge \cdots \wedge \tilde{\psi}_{p_N}(\vec{r}_N), \tag{49}$$

where $\tilde{\psi}_{p_1}(\vec{r}_1) \wedge \cdots \wedge \tilde{\psi}_{p_N}(\vec{r}_N)$ designates the normalized antisymmetrized tensor product of $N$ orbitals, i.e. a Slater determinant. In this approximation, the vacuum contributions are taken into account at the mean-field level. Indeed, using the rewriting of the electron-positron Hamiltonian in Eq. (33), we have

$$E_N^{\text{npvp}} = \langle \Psi_+ | \tilde{\hat{T}}_{\text{D}} + \tilde{\hat{W}} + \tilde{\hat{V}} + \tilde{\hat{V}}^{\text{vp}} | \Psi_+ \rangle + \tilde{E}_0, \tag{50}$$

which includes the vacuum-polarization potential operator [Eq. (38)] and the vacuum energy [Eq. (41)].

The common no-pair (np) approximation corresponds to additionally neglecting all vacuum contributions

$$E_N^{\text{np}} = \min_{|\Psi_+\rangle \in \tilde{\mathcal{H}}^{(N,0)}} \langle \Psi_+ | \tilde{\hat{T}}_{\text{D}} + \tilde{\hat{W}} + \tilde{\hat{V}} | \Psi_+ \rangle, \tag{51}$$

where we use now the Hamiltonian written with normal ordering with respect to a floating vacuum state $|\tilde{0}\rangle$. The no-pair approximation with optimized orbitals is analogous to the complete-active-space self-consistent-field method of quantum chemistry in which the wave function is expanded in the Hilbert space spanned by only a subset of orbitals (the equivalent of the PS set) and the orbitals are optimized by performing rotations with the complementary subset of orbitals (the equivalent of the NS set).

Note that in Eq. (46) or (51) one can minimize with respect to the projector $\tilde{\hat{P}}_+$ thanks to the use of the Fock-space normal-ordered electron-positron Hamiltonian. If one starts instead with the configuration-space Dirac-Coulomb or Dirac-Coulomb-Breit Hamiltonian, the same $E_N^{\text{np}}$ can be obtained but using instead a minmax principle in which the energy is maximized with respect to the projector (see Refs. [23, 50–52]).

# 3 Density-functional theory based on effective quantum electro-dynamics

We now formulate a RDFT based on the electron-positron Hamiltonian in Eq. (7). We will consider the simplest case of functionals of only the opposite charge density $n(\vec{r}) = \langle \Psi | \hat{n}(\vec{r}) | \Psi \rangle$, which is appropriate for closed-shell systems. More generally, one could consider functionals depending also on the opposite charge current $\vec{j}(\vec{r}) = \langle \Psi | \hat{\vec{j}}(\vec{r}) | \Psi \rangle$ with $\hat{\vec{j}}(\vec{r}) = \text{Tr}[c \vec{\alpha} \, \hat{\mathbf{n}}(\vec{r})]$. Even more generally, one could consider functionals of the local density matrix $\mathbf{n}(\vec{r}) = \langle \Psi | \hat{\mathbf{n}}(\vec{r}) | \Psi \rangle$, as proposed in Ref. [13]. For simplicity, in the following, the opposite charge density and opposite charge current will be referred to as charge density and charge current.

## 3.1 Kohn-Sham scheme

Using the constrained-search formalism [32, 33], we define the universal density functional $F[n]$ for $N$-representable charge densities $n \in \mathcal{D}_N$, i.e. charge densities that come from a state $|\Psi\rangle \in \mathcal{H}_N$,

$$F[n] \quad = \quad \min_{|\Psi\rangle \in \mathcal{H}_N(n)} \langle \Psi | \hat{T}_{\text{D}} + \hat{W} | \Psi \rangle = \langle \Psi[n] | \hat{T}_{\text{D}} + \hat{W} | \Psi[n] \rangle, \tag{52}$$

where $\mathcal{H}_N(n)$ is the set of states $|\Psi\rangle \in \mathcal{H}_N$ constrained to yield the charge density $n$, and $|\Psi[n]\rangle$ designates a minimizing state. A $N$-representable charge density must of course contain a net total amount of $N$ negative charges, i.e. $\int n(\vec{r})\text{d}\vec{r} = N$, but other than that the set of $N$-representable charge densities $\mathcal{D}_N$ is a priori unknown. This is unlike the non-relativistic case for which the mathematical set of $N$-representable densities is explicitly known [33]. The $N$-negative-charge ground-state energy can then be written as

$$E_N \quad = \quad \min_{n \in \mathcal{D}_N} \left[ F[n] + \int v(\vec{r}) \, n(\vec{r}) \, \text{d}\vec{r} \right]. \tag{53}$$

Note that, in the special case $N = 0$ we obtain the correlated vacuum energy of Sec. 2.4. Also, as already indicated, we can allow for negative $N$ to describe the case of $N$ positive charges.

To setup a KS scheme [53], we decompose $F[n]$ as

$$F[n] = T_{\text{s}}[n] + E_{\text{Hxc}}[n], \tag{54}$$

where $T_{\text{s}}[n]$ is the non-interacting kinetic + rest-mass density functional

$$T_{\text{s}}[n] = \min_{|\Phi\rangle \in \tilde{\mathcal{S}}^{(N,0)}(n)} \langle \Phi | \hat{T}_{\text{D}} | \Phi \rangle = \langle \Phi[n] | \hat{T}_{\text{D}} | \Phi[n] \rangle, \tag{55}$$

where the minimization is over the set $\tilde{\mathcal{S}}^{(N,0)}(n)$ of single-determinant states $|\Phi\rangle = \hat{\tilde{b}}_1^\dagger \hat{\tilde{b}}_2^\dagger \cdots \hat{\tilde{b}}_N^\dagger | \tilde{0} \rangle$ with a fixed number of electrons $N$ with respect to a floating vacuum state and yielding the charge density $n$, and $E_{\text{Hxc}}[n]$ is the Hartree-exchange-correlation density functional. The minimizing state (that we will assume unique up to a phase factor for simplicity) is the KS single-determinant state $|\Phi[n]\rangle$. Note that in Eq. (55) we have tacitly assumed that any $N$-representable charge density $n$ can be represented by a single-determinant state $|\Phi\rangle$. For the non-relativistic theory, this can be proved to be true by explicitly constructing a single determinant yielding any given $N$-representable density [33, 54, 55]. This proof does not apply to the present relativistic theory due to the more complicated form of the charge density $n(\vec{r})$ which includes the vacuum-polarization contribution [see Eqs. (62) and (63)].

In fact, due to the vacuum-polarization contribution, the charge density $n(\vec{r})$ may not generally have the same sign at all spatial points. This is particularly obvious for the case $N = 0$: the charge density integrates to zero $\int n(\vec{r})\mathrm{d}\vec{r} = 0$ and thus necessarily changes sign. Whether the proofs of Refs. [33,54,55] can be generalized to the relativistic case is an open question. We can then write the ground-state energy as

$$E_N = \min_{|\Phi\rangle \in \tilde{\mathcal{S}}^{(N,0)}} \left[ \langle \Phi | \hat{T}_\mathrm{D} + \hat{V} | \Phi \rangle + E_\mathrm{Hxc}[n_{|\Phi\rangle}] \right], \tag{56}$$

where $\tilde{\mathcal{S}}^{(N,0)}$ is the set of single-determinant states with a fixed number of electrons $N$ with respect to a floating vacuum state. Note that, contrary to a general $N$-negative-charge state in Eq. (45), we can associate a wave function to a single-determinant state, i.e. $\Phi(\vec{r}_1, \vec{r}_2, ..., \vec{r}_N) = \tilde{\psi}_1(\vec{r}_1) \wedge \cdots \wedge \tilde{\psi}_N(\vec{r}_N)$.

More explicitly, the expression of the energy in terms of the orbitals $\{\tilde{\psi}_p\}$ is

$$E_N[\{\tilde{\psi}_p\}] = \int \mathrm{Tr}[\mathbf{D}(\vec{r})\,\mathbf{n}_1^\mathrm{KS}(\vec{r},\vec{r}\,')]_{\vec{r}\,'=\vec{r}}\,\mathrm{d}\vec{r} + \int v(\vec{r})\,n(\vec{r})\,\mathrm{d}\vec{r} + E_\mathrm{Hxc}[n], \tag{57}$$

with the KS one-particle density matrix

$$\mathbf{n}_1^\mathrm{KS}(\vec{r},\vec{r}\,') = \tilde{\mathbf{n}}_1^\mathrm{KS}(\vec{r},\vec{r}\,') + \tilde{\mathbf{n}}_1^\mathrm{vp}(\vec{r},\vec{r}\,'), \tag{58}$$

which includes the contribution from the electronic occupied orbitals

$$\tilde{\mathbf{n}}_1^\mathrm{KS}(\vec{r},\vec{r}\,') = \sum_{i=1}^N \tilde{\psi}_i(\vec{r})\tilde{\psi}_i^\dagger(\vec{r}\,'), \tag{59}$$

and from the vacuum polarization [see Eq. (31)]

$$\tilde{\mathbf{n}}_1^\mathrm{vp}(\vec{r},\vec{r}\,') = \sum_{p\in\mathrm{NS}} \tilde{\psi}_p(\vec{r})\tilde{\psi}_p^\dagger(\vec{r}\,') - \sum_{p\in\mathrm{NS}} \psi_p(\vec{r})\psi_p^\dagger(\vec{r}\,'), \tag{60}$$

and with the corresponding charge density $n(r) = \mathrm{Tr}[\mathbf{n}_1^\mathrm{KS}(\vec{r},\vec{r})]$. Taking the functional derivative of $E_N[\{\psi_p\}]$ with respect to $\tilde{\psi}_p^\dagger(\vec{r})$ with the orbital orthonormalization constraints, we arrive at the KS equations

$$(\mathbf{D}(\vec{r}) + v(\vec{r}) + v_\mathrm{Hxc}(\vec{r}))\,\tilde{\psi}_p(\vec{r}) = \tilde{\varepsilon}_p \tilde{\psi}_p(\vec{r}), \tag{61}$$

where $v_\mathrm{Hxc}(\vec{r}) = \delta E_\mathrm{Hxc}[n]/\delta n(\vec{r})$ is the Hartree-exchange-correlation potential (assuming a form of differentiability of the functional $E_\mathrm{Hxc}[n]$) and $\tilde{\varepsilon}_p$ are the KS orbital energies. The KS equations must be solved self-consistently with the density

$$n(\vec{r}) = \sum_{i=1}^N \tilde{\psi}_i^\dagger(\vec{r})\tilde{\psi}_i(\vec{r}) + \tilde{n}^\mathrm{vp}(\vec{r}), \tag{62}$$

where the vacuum-polarization density is

$$\begin{aligned}
\tilde{n}^\mathrm{vp}(\vec{r}) &= \sum_{p\in\mathrm{NS}} \tilde{\psi}_p^\dagger(\vec{r})\tilde{\psi}_p(\vec{r}) - \sum_{p\in\mathrm{NS}} \psi_p^\dagger(\vec{r})\psi_p(\vec{r}) \\
&= \frac{1}{2}\left( \sum_{p\in\mathrm{NS}} \tilde{\psi}_p^\dagger(\vec{r})\tilde{\psi}_p(\vec{r}) - \sum_{p\in\mathrm{PS}} \tilde{\psi}_p^\dagger(\vec{r})\tilde{\psi}_p(\vec{r}) \right),
\end{aligned} \tag{63}$$

where the last equality follows from Eqs. (168) and (172) (see also Ref. [56]). Equations (61)-(63) have a similar form as for the KS scheme based on renormalized QED [7–10] except that we did not take into account any renormalization counterterms and that the present functional $E_\mathrm{Hxc}[n]$ is associated with the effective Coulomb or Coulomb+Breit two-particle interaction. The fact that $E_\mathrm{Hxc}[n]$ is a functional of the density makes the potential $v_\mathrm{Hxc}(\vec{r})$ local in space and diagonal in terms of spinor indices. This is unlike in HF theory where the corresponding potential would be both nonlocal in space and non-diagonal in terms of spinor indices.

## 3.2 Hartree-exchange-correlation density functional

The Hartree-exchange-correlation density functional $E_{\mathrm{Hxc}}[n]$ can be decomposed as

$$E_{\mathrm{Hxc}}[n] = E_{\mathrm{Hx}}[n] + E_{\mathrm{c}}[n], \tag{64}$$

where $E_{\mathrm{Hx}}[n]$ is the Hartree-exchange energy encompassing all first-order terms in the two-particle interaction

$$E_{\mathrm{Hx}}[n] = \langle\Phi[n]|\hat{W}|\Phi[n]\rangle = \frac{1}{2}\iint \mathrm{Tr}[\mathbf{w}(\vec{r}_1,\vec{r}_2)\mathbf{n}_2^{\mathrm{KS}}(\vec{r}_1,\vec{r}_2)]\mathrm{d}\vec{r}_1\mathrm{d}\vec{r}_2, \tag{65}$$

with the KS pair-density matrix $\mathbf{n}_2^{\mathrm{KS}}(\vec{r}_1,\vec{r}_2) = \langle\Phi[n]|\hat{\mathbf{n}}_2(\vec{r}_1,\vec{r}_2)|\Phi[n]\rangle$, and $E_{\mathrm{c}}[n]$ is the correlation energy. The Hartree-exchange energy can be written more explicitly as

$$E_{\mathrm{Hx}}[n] = \tilde{E}_{\mathrm{Hx}}[n] + \tilde{E}_{\mathrm{Hx}}^{\mathrm{vp}}[n], \tag{66}$$

where $\tilde{E}_{\mathrm{Hx}}[n]$ is the main contribution

$$\tilde{E}_{\mathrm{Hx}}[n] = \frac{1}{2}\iint \mathrm{Tr}[\mathbf{w}(\vec{r}_1,\vec{r}_2)\tilde{\mathbf{n}}_2^{\mathrm{KS}}(\vec{r}_1,\vec{r}_2)]\mathrm{d}\vec{r}_1\mathrm{d}\vec{r}_2, \tag{67}$$

depending on the part of the KS pair-density matrix coming from the electronic occupied orbitals

$$\tilde{n}_{2,\rho\nu\sigma\tau}^{\mathrm{KS}}(\vec{r}_1,\vec{r}_2) = \tilde{n}_{1,\nu\tau}^{\mathrm{KS}}(\vec{r}_2,\vec{r}_2)\tilde{n}_{1,\rho\sigma}^{\mathrm{KS}}(\vec{r}_1,\vec{r}_1) - \tilde{n}_{1,\rho\tau}^{\mathrm{KS}}(\vec{r}_1,\vec{r}_2)\tilde{n}_{1,\nu\sigma}^{\mathrm{KS}}(\vec{r}_2,\vec{r}_1), \tag{68}$$

and $\tilde{E}_{\mathrm{Hx}}^{\mathrm{vp}}[n]$ is the vacuum-polarization contribution

$$
\begin{aligned}
\tilde{E}_{\mathrm{Hx}}^{\mathrm{vp}}[n] = {} & \int \mathrm{Tr}[\tilde{\mathbf{v}}_{\mathrm{H}}^{\mathrm{vp}}(\vec{r})\tilde{\mathbf{n}}_1^{\mathrm{KS}}(\vec{r},\vec{r})]\mathrm{d}\vec{r} + \iint \mathrm{Tr}[\tilde{\mathbf{v}}_{\mathrm{x}}^{\mathrm{vp}}(\vec{r}_1,\vec{r}_2)\tilde{\mathbf{n}}_1^{\mathrm{KS}}(\vec{r}_1,\vec{r}_2)]\mathrm{d}\vec{r}_1\mathrm{d}\vec{r}_2 \\
& + \frac{1}{2}\iint \mathrm{Tr}[\mathbf{w}(\vec{r}_1,\vec{r}_2)\tilde{\mathbf{n}}_2^{\mathrm{vp}}(\vec{r}_1,\vec{r}_2)]\mathrm{d}\vec{r}_1\mathrm{d}\vec{r}_2,
\end{aligned}
\tag{69}
$$

where the vacuum-polarization potentials $\tilde{\mathbf{v}}_{\mathrm{H}}^{\mathrm{vp}}(\vec{r})$ and $\tilde{\mathbf{v}}_{\mathrm{x}}^{\mathrm{vp}}(\vec{r}_1,\vec{r}_2)$ were defined after Eqs. (39) and (40), respectively, and the vacuum-polarization pair-density matrix $\tilde{\mathbf{n}}_2^{\mathrm{vp}}(\vec{r}_1,\vec{r}_2)$ was defined in Eq. (32).

We can further decompose the functional $E_{\mathrm{Hx}}[n]$ as

$$E_{\mathrm{Hx}}[n] = E_{\mathrm{H}}[n] + E_{\mathrm{x}}[n], \tag{70}$$

where the Hartree functional $E_{\mathrm{H}}[n]$ collects all direct terms and the exchange functional $E_{\mathrm{x}}[n]$ collects all exchange terms. The expression of the Hartree functional is

$$E_{\mathrm{H}}[n] = \tilde{E}_{\mathrm{H}}[n] + \tilde{E}_{\mathrm{H}}^{\mathrm{vp}}[n], \tag{71}$$

with

$$\tilde{E}_{\mathrm{H}}[n] = \frac{1}{2}\iint \mathrm{Tr}[\mathbf{w}(\vec{r}_1,\vec{r}_2)\tilde{\mathbf{n}}_{2,\mathrm{H}}^{\mathrm{KS}}(\vec{r}_1,\vec{r}_2)]\mathrm{d}\vec{r}_1\mathrm{d}\vec{r}_2, \tag{72}$$

where $\tilde{\mathbf{n}}_{2,\mathrm{H}}^{\mathrm{KS}}(\vec{r}_1,\vec{r}_2)$ is the Hartree contribution to $\tilde{\mathbf{n}}_2^{\mathrm{KS}}(\vec{r}_1,\vec{r}_2)$ [the first term in the right-hand side of Eq. (68)], and

$$\tilde{E}_{\mathrm{H}}^{\mathrm{vp}}[n] = \int \mathrm{Tr}[\tilde{\mathbf{v}}_{\mathrm{H}}^{\mathrm{vp}}(\vec{r})\tilde{\mathbf{n}}_1^{\mathrm{KS}}(\vec{r},\vec{r})]\mathrm{d}\vec{r} + \frac{1}{2}\iint \mathrm{Tr}[\mathbf{w}(\vec{r}_1,\vec{r}_2)\tilde{\mathbf{n}}_{2,\mathrm{H}}^{\mathrm{vp}}(\vec{r}_1,\vec{r}_2)]\mathrm{d}\vec{r}_1\mathrm{d}\vec{r}_2, \tag{73}$$

where $\tilde{\mathbf{n}}_{2,\mathrm{H}}^{\mathrm{vp}}(\vec{r}_1,\vec{r}_2)$ is the Hartree contribution to $\tilde{\mathbf{n}}_2^{\mathrm{vp}}(\vec{r}_1,\vec{r}_2)$ [the first term in the right-hand side of Eq. (32)]. Similarly, the expression of the exchange functional is

$$E_{\mathrm{x}}[n] = \tilde{E}_{\mathrm{x}}[n] + \tilde{E}_{\mathrm{x}}^{\mathrm{vp}}[n], \tag{74}$$

with

$$\tilde{E}_{\mathrm{x}}[n] = \frac{1}{2}\iint \mathrm{Tr}[\mathbf{w}(\vec{r}_1,\vec{r}_2)\tilde{\mathbf{n}}_{2,\mathrm{x}}^{\mathrm{KS}}(\vec{r}_1,\vec{r}_2)]\mathrm{d}\vec{r}_1\mathrm{d}\vec{r}_2, \tag{75}$$

where $\tilde{\mathbf{n}}_{2,\mathrm{x}}^{\mathrm{KS}}(\vec{r}_1,\vec{r}_2)$ is the exchange contribution to $\tilde{\mathbf{n}}_2^{\mathrm{KS}}(\vec{r}_1,\vec{r}_2)$ [the second term in the right-hand side of Eq. (68)], and

$$\tilde{E}_{\mathrm{x}}^{\mathrm{vp}}[n] = \iint \mathrm{Tr}[\tilde{\mathbf{v}}_{\mathrm{x}}^{\mathrm{vp}}(\vec{r}_1,\vec{r}_2)\tilde{\mathbf{n}}_1^{\mathrm{KS}}(\vec{r}_1,\vec{r}_2)]\mathrm{d}\vec{r}_1\mathrm{d}\vec{r}_2 + \frac{1}{2}\iint \mathrm{Tr}[\mathbf{w}(\vec{r}_1,\vec{r}_2)\tilde{\mathbf{n}}_{2,\mathrm{x}}^{\mathrm{vp}}(\vec{r}_1,\vec{r}_2)]\mathrm{d}\vec{r}_1\mathrm{d}\vec{r}_2, \tag{76}$$

where $\tilde{\mathbf{n}}_{2,\mathrm{x}}^{\mathrm{vp}}(\vec{r}_1,\vec{r}_2)$ is the exchange contribution to $\tilde{\mathbf{n}}_2^{\mathrm{vp}}(\vec{r}_1,\vec{r}_2)$ [the second term in the right-hand side of Eq. (32)].

The Hartree energy can also be more compactly written as a sum of Coulomb and Breit contributions

$$E_{\mathrm{H}}[n] = E_{\mathrm{H}}^{\mathrm{C}}[n] + E_{\mathrm{H}}^{\mathrm{B}}[n], \tag{77}$$

where the Coulomb contribution has the same form as in non-relativistic DFT

$$E_{\mathrm{H}}^{\mathrm{C}}[n] = \frac{1}{2}\iint w(r_{12})n(\vec{r}_1)n(\vec{r}_2)\mathrm{d}\vec{r}_1\mathrm{d}\vec{r}_2, \tag{78}$$

involving the charge density $n(\vec{r})$ [Eq. (62)], and the Breit contribution has the form

$$E_{\mathrm{H}}^{\mathrm{B}}[n] = -\frac{1}{4c^2}\iint w(r_{12})\left[\vec{j}(\vec{r}_1)\cdot\vec{j}(\vec{r}_2) + \frac{\vec{j}(\vec{r}_1)\cdot\vec{r}_{12}\ \vec{j}(\vec{r}_2)\cdot\vec{r}_{12}}{r_{12}^2}\right]\mathrm{d}\vec{r}_1\mathrm{d}\vec{r}_2, \tag{79}$$

involving the KS charge current density $\vec{j}(\vec{r})$

$$\vec{j}(\vec{r}) = \mathrm{Tr}[c\vec{\alpha}\ \mathbf{n}_1^{\mathrm{KS}}(\vec{r},\vec{r})] = c\sum_{i=1}^{N}\tilde{\psi}_i^{\dagger}(\vec{r})\vec{\alpha}\tilde{\psi}_i(\vec{r}) + \vec{j}^{\mathrm{vp}}(\vec{r}), \tag{80}$$

where $\vec{j}^{\mathrm{vp}}(\vec{r})$ is the vacuum-polarization current density

$$\vec{j}^{\mathrm{vp}}(\vec{r}) = c\left[\sum_{p\in\mathrm{NS}}\tilde{\psi}_p^{\dagger}(\vec{r})\vec{\alpha}\tilde{\psi}_p(\vec{r}) - \sum_{p\in\mathrm{NS}}\psi_p^{\dagger}(\vec{r})\vec{\alpha}\psi_p(\vec{r})\right]. \tag{81}$$

Since we did not consider any vector potential in the KS equations [Eq. (61)], the KS Hamiltonian has time-reversal symmetry and the KS orbitals $\{\tilde{\psi}_p\}$ come in degenerate Kramers pairs (see, e.g., Ref. [23]) with opposite current densities, and similarly for the orbitals $\{\psi_p\}$ of the free Dirac equation. It seems then reasonable to conclude that the vacuum-polarization current density $\vec{j}^{\mathrm{vp}}(\vec{r})$ vanishes in the present context, glossing over the fact that each sum in Eq. (81) is infinite. Moreover, for closed-shell systems, the contribution to the charge current density $\vec{j}(\vec{r})$ coming from the occupied electronic states in Eq. (80) vanishes as well, and there is no Breit contribution to the Hartree energy. For open-shell systems, the charge current density does not vanish and there is a Breit contribution to the Hartree energy. Since

the charge current density $\vec{j}(\vec{r})$ is only an implicit functional of the charge density via the KS orbitals, the calculation of the Breit contribution to the Hartree potential would require to use the optimized-effective-potential method (see, e.g., Ref. [57]). A simpler alternative is to switch to functionals depending also explicitly on the charge current density $\vec{j}(\vec{r})$.

The correlation functional $E_c[n]$ is conveniently expressed with the adiabatic-connection approach [58–60] which can be straightforwardly generalized to the present relativistic theory. For this, we define an universal density functional similarly to Eq. (52) but depending on a coupling constant $\lambda \in [0, +\infty[$

$$F^\lambda[n] = \min_{|\Psi\rangle \in \mathcal{H}_N(n)} \langle \Psi | \hat{T}_D + \lambda \hat{W} | \Psi \rangle = \langle \Psi^\lambda[n] | \hat{T}_D + \lambda \hat{W} | \Psi^\lambda[n] \rangle, \tag{82}$$

where $|\Psi^\lambda[n]\rangle$ denotes a minimizing state. This functional can be decomposed as

$$F^\lambda[n] = T_s[n] + \lambda E_{Hx}[n] + E_c^\lambda[n], \tag{83}$$

where the $\lambda$-dependent correlation contribution is

$$E_c^\lambda[n] = \langle \Psi^\lambda[n] | \hat{T}_D + \lambda \hat{W} | \Psi^\lambda[n] \rangle - \langle \Phi[n] | \hat{T}_D + \lambda \hat{W} | \Phi[n] \rangle. \tag{84}$$

We will assume that $F^\lambda[n]$ is of class $C^1$ as a function of $\lambda$ for $\lambda \in [0, 1]$ and that $F^{\lambda=0}[\rho] = T_s[\rho]$ (which should be valid when the KS single-determinant state $|\Phi[n]\rangle$ is non-degenerate). Taking the derivative of Eq. (84) with respect to $\lambda$ and using the Hellmann-Feynman theorem for the state $|\Psi^\lambda[n]\rangle$, we obtain

$$\frac{\partial E_c^\lambda[n]}{\partial \lambda} = \langle \Psi^\lambda[n] | \hat{W} | \Psi^\lambda[n] \rangle - \langle \Phi[n] | \hat{W} | \Phi[n] \rangle. \tag{85}$$

Integrating over $\lambda$ from 0 to 1, and using $E_c^{\lambda=1}[n] = E_c[n]$ and $E_c^{\lambda=0}[n] = 0$, we arrive at the adiabatic-connection formula for the correlation functional

$$\begin{aligned} E_c[n] &= \int_0^1 d\lambda \, \langle \Psi^\lambda[n] | \hat{W} | \Psi^\lambda[n] \rangle - \langle \Phi[n] | \hat{W} | \Phi[n] \rangle \\ &= \frac{1}{2} \int_0^1 d\lambda \iint \text{Tr}[\mathbf{w}(\vec{r}_1, \vec{r}_2) \mathbf{n}_{2,c}^\lambda(\vec{r}_1, \vec{r}_2)] d\vec{r}_1 d\vec{r}_2, \end{aligned} \tag{86}$$

with the correlation contribution to the $\lambda$-dependent pair-density matrix $\mathbf{n}_{2,c}^\lambda(\vec{r}_1, \vec{r}_2) = \langle \Psi^\lambda[n] | \hat{\mathbf{n}}_2(\vec{r}_1, \vec{r}_2) | \Psi^\lambda[n] \rangle - \mathbf{n}_2^{KS}(\vec{r}_1, \vec{r}_2)$. More explicitly, the correlation functional has the expression

$$E_c[n] = \tilde{E}_c[n] + \tilde{E}_c^{vp}[n], \tag{87}$$

where $\tilde{E}_c[n]$ is the main contribution

$$\tilde{E}_c[n] = \frac{1}{2} \int_0^1 d\lambda \iint \text{Tr}[\mathbf{w}(\vec{r}_1, \vec{r}_2) \tilde{\mathbf{n}}_{2,c}^\lambda(\vec{r}_1, \vec{r}_2)] d\vec{r}_1 d\vec{r}_2, \tag{88}$$

with $\tilde{\mathbf{n}}_{2,c}^\lambda(\vec{r}_1, \vec{r}_2) = \langle \Psi^\lambda[n] | \hat{\tilde{\mathbf{n}}}_2(\vec{r}_1, \vec{r}_2) | \Psi^\lambda[n] \rangle - \tilde{\mathbf{n}}_2^{KS}(\vec{r}_1, \vec{r}_2)$, and $\tilde{E}_c^{vp}[n]$ is the vacuum-polarization contribution coming from the variation of the one-particle density matrix with $\lambda$

$$\tilde{E}_c^{vp}[n] = \int_0^1 d\lambda \int \text{Tr}[\tilde{\mathbf{v}}_H^{vp}(\vec{r}) \tilde{\mathbf{n}}_{1,c}^\lambda(\vec{r}, \vec{r})] d\vec{r} + \int_0^1 d\lambda \iint \text{Tr}[\tilde{\mathbf{v}}_x^{vp}(\vec{r}_1, \vec{r}_2) \tilde{\mathbf{n}}_{1,c}^\lambda(\vec{r}_1, \vec{r}_2)] d\vec{r}_1 d\vec{r}_2, \tag{89}$$

with $\tilde{\mathbf{n}}_{1,c}^{\lambda}(\vec{r}_1, \vec{r}_2) = \langle \Psi^{\lambda}[n] | \hat{\tilde{\mathbf{n}}}_1(\vec{r}_1, \vec{r}_2) | \Psi^{\lambda}[n] \rangle - \tilde{\mathbf{n}}_1^{\text{KS}}(\vec{r}_1, \vec{r}_2)$. Note that both $\tilde{\mathbf{n}}_{2,c}^{\lambda}(\vec{r}_1, \vec{r}_2)$ and $\tilde{\mathbf{n}}_{1,c}^{\lambda}(\vec{r}_1, \vec{r}_2)$ include contributions from orbitals $\tilde{\psi}_p$ with $p \in \text{NS}$, which generate vacuum contributions to the correlation energy beyond first order in the two-particle interaction.

Mirroring the decomposition of the energy functional $E_{\text{Hxc}}[n]$ into Hartree, exchange, and correlation contributions, the associated potential in Eq. (61) has of course a similar decomposition

$$v_{\text{Hxc}}(\mathbf{r}) = v_{\text{H}}(\mathbf{r}) + v_{\text{x}}(\mathbf{r}) + v_{\text{c}}(\mathbf{r}), \tag{90}$$

and each potential is itself a sum of a main contribution and a vacuum-polarization contribution. Note in particular that the vacuum-polarization contributions in the Hartree and exchange potentials are both local in space and diagonal in terms of spinor indices and thus are not identical to the vacuum-polarization potentials $\tilde{v}_{\text{H}}^{\text{vp}}(\vec{r})$ and $\tilde{v}_{\text{x}}^{\text{vp}}(\vec{r}_1, \vec{r}_2)$ defined after Eqs. (39) and (40), respectively. The latter potentials are the vacuum-polarization potentials that would be directly involved in HF theory. We leave for future work the study of the properties of the potentials in Eq. (90).

### 3.3 No-pair approximation

In the npvp approximation introduced in Eq. (46), the universal density functional becomes

$$F^{\text{npvp}}[n] = \min_{|\Psi_+\rangle \in \tilde{\mathcal{H}}^{(N,0)}(n)} \langle \Psi_+ | \hat{T}_{\text{D}} + \hat{W} | \Psi_+ \rangle, \tag{91}$$

where $\tilde{\mathcal{H}}^{(N,0)}(n)$ is the set of states in $\tilde{\mathcal{H}}^{(N,0)}$ yielding the charge density $n$. In this approximation, the definition of $T_s[n]$ in Eq. (55) is left unchanged and consequently the KS determinant state $|\Phi[n]\rangle$ and the Hartree and exchange functionals $E_{\text{H}}[n]$ and $E_{\text{x}}[n]$ are also left unchanged. We thus have the decomposition

$$F^{\text{npvp}}[n] = T_s[n] + E_{\text{Hx}}[n] + E_c^{\text{npvp}}[n], \tag{92}$$

where $E_c^{\text{npvp}}[n]$ is the new correlation functional in this approximation. In this npvp KS scheme, the ground-state energy is then obtained as

$$E_N^{\text{npvp}} = \min_{|\Phi\rangle \in \tilde{\mathcal{S}}^{(N,0)}} \left[ \langle \Phi | \hat{T}_{\text{D}} + \hat{V} | \Phi \rangle + E_{\text{Hx}}[n_{|\Phi\rangle}] + E_c^{\text{npvp}}[n_{|\Phi\rangle}] \right]. \tag{93}$$

Hence, this approximation affects only the correlation functional, namely $E_c^{\text{npvp}}[n]$ has the same expression as $E_c[n]$ but in Eqs. (88) and (89) $\tilde{\mathbf{n}}_{2,c}^{\lambda}(\vec{r}_1, \vec{r}_2)$ and $\tilde{\mathbf{n}}_{1,c}^{\lambda}(\vec{r}_1, \vec{r}_2)$ are now calculated with a state $|\Psi_+^{\lambda}[n]\rangle \in \tilde{\mathcal{H}}^{(N,0)}(n)$ and thus do not contain any contributions coming from orbitals $\tilde{\psi}_p$ with $p \in \text{NS}$. However, vacuum contributions are still included at the mean-field level with the potentials $\tilde{v}_{\text{H}}^{\text{vp}}(\vec{r})$ and $\tilde{v}_{\text{x}}^{\text{vp}}(\vec{r}_1, \vec{r}_2)$.

In the more common no-pair approximation of Eq. (51), the universal functional is defined as

$$F^{\text{np}}[n] = \min_{|\Psi_+\rangle \in \tilde{\mathcal{H}}^{(N,0)}(n)} \langle \Psi_+ | \hat{\tilde{T}}_{\text{D}} + \hat{\tilde{W}} | \Psi_+ \rangle, \tag{94}$$

where we use now the operators written with normal ordering with respect to a floating vacuum state $|\tilde{0}\rangle$, and the non-interacting kinetic + rest-mass density functional is defined as

$$T_s^{\text{np}}[n] = \min_{|\Phi\rangle \in \tilde{\mathcal{S}}^{(N,0)}(n)} \langle \Phi | \hat{\tilde{T}}_{\text{D}} | \Phi \rangle = \langle \Phi^{\text{np}}[n] | \hat{\tilde{T}}_{\text{D}} | \Phi^{\text{np}}[n] \rangle, \tag{95}$$

where $|\Phi^{\mathrm{np}}[n]\rangle$ is the KS determinant state in this approximation (again, assumed to be unique up to a phase factor for simplicity). The functional $F^{\mathrm{np}}[n]$ can then be decomposed as

$$F^{\mathrm{np}}[n] = T_s^{\mathrm{np}}[n] + E_{\mathrm{Hx}}^{\mathrm{np}}[n] + E_c^{\mathrm{np}}[n], \tag{96}$$

where $E_{\mathrm{Hx}}^{\mathrm{np}}[n]$ is the no-pair Hartree-exchange functional

$$E_{\mathrm{Hx}}^{\mathrm{np}}[n] = \langle \Phi^{\mathrm{np}}[n]|\hat{\bar{W}}|\Phi^{\mathrm{np}}[n]\rangle = \frac{1}{2}\iint \mathrm{Tr}[\mathbf{w}(\vec{r}_1,\vec{r}_2)\tilde{\mathbf{n}}_2^{\mathrm{KS,np}}(\vec{r}_1,\vec{r}_2)]\mathrm{d}\vec{r}_1\mathrm{d}\vec{r}_2, \tag{97}$$

with the no-pair KS pair-density matrix $\tilde{\mathbf{n}}_2^{\mathrm{KS,np}}(\vec{r}_1,\vec{r}_2) = \langle \Phi^{\mathrm{np}}[n]|\hat{\tilde{\mathbf{n}}}_2(\vec{r}_1,\vec{r}_2)|\Phi^{\mathrm{np}}[n]\rangle$ (which, as before, can be trivially separated into Hartree and exchange contributions), and $E_c^{\mathrm{np}}[n]$ is the no-pair correlation functional

$$E_c^{\mathrm{np}}[n] = \frac{1}{2}\int_0^1 \mathrm{d}\lambda \iint \mathrm{Tr}[\mathbf{w}(\vec{r}_1,\vec{r}_2)\tilde{\mathbf{n}}_{2,c}^{\lambda,\mathrm{np}}(\vec{r}_1,\vec{r}_2)]\mathrm{d}\vec{r}_1\mathrm{d}\vec{r}_2, \tag{98}$$

with $\tilde{\mathbf{n}}_{2,c}^{\lambda,\mathrm{np}}(\vec{r}_1,\vec{r}_2) = \langle \Psi_+^{\lambda}[n]|\hat{\tilde{\mathbf{n}}}_2(\vec{r}_1,\vec{r}_2)|\Psi_+^{\lambda}[n]\rangle - \tilde{\mathbf{n}}_2^{\mathrm{KS,np}}(\vec{r}_1,\vec{r}_2)$ and $|\Psi_+^{\lambda}[n]\rangle$ is a $\lambda$-dependent no-pair minimizing state for the charge density $n$. Finally, the no-pair ground-state energy is obtained as

$$E_N^{\mathrm{np}} = \min_{|\Phi\rangle \in \tilde{\mathcal{S}}^{(N,0)}} \left[ \langle \Phi|\hat{\bar{T}}_{\mathrm{D}} + \hat{\bar{V}}|\Phi\rangle + E_{\mathrm{Hx}}^{\mathrm{np}}[n_{|\Phi\rangle}] + E_c^{\mathrm{np}}[n_{|\Phi\rangle}] \right], \tag{99}$$

and the no-pair charge density is simply $n(\vec{r}) = \sum_{i=1}^N \tilde{\psi}_i^\dagger(\vec{r})\tilde{\psi}_i(\vec{r})$.

This constitutes a no-pair KS RDFT with well-defined universal exchange and correlation functionals $E_x^{\mathrm{np}}[n]$ and $E_c^{\mathrm{np}}[n]$. This contrasts with the RDFT based on the relativistic extension of the Hohenberg-Kohn theorem of Refs. [7–10] for which the no-pair approximation is only introduced a posteriori without giving an unambiguous definition of the involved functionals. Indeed, the no-pair approximation involves the projector $\hat{\bar{P}}_+$ onto the subspace of electronic states [Eq. (48)] which depends on the separation of the orbitals into PS and NS sets, and therefore depends on the potential used to generate these orbitals. If the projector is applied to the Hamiltonian, the whole resulting projected Hamiltonian is thus dependent on this potential, and one cannot isolate, as normally done in DFT, an universal part of the Hamiltonian, and one thus cannot define universal density functionals. In the present work, instead of thinking of the projector $\hat{\bar{P}}_+$ as being applied to the Hamiltonian, we equivalently think of the projector as being applied to the state, i.e. $|\Psi_+\rangle = \hat{\bar{P}}_+|\Psi\rangle$, and optimize the projector simultaneously with the state $|\Psi\rangle$. In this way, we can introduce universal density functionals, similarly to non-relativistic DFT, defined such that for a given density a constrained-search optimization in Eq. (94) or (95) of the projected state $|\Psi_+\rangle$ determines alone the optimal projector without the need of pre-choosing a particular potential, at least for systems for which orbitals can be unambiguously separated into PS and NS sets. The same view can be taken in the configuration-space approach using a minmax principle [52].

## 3.4 Exact properties of the density functionals

**Charge-conjugation symmetry**

A state $|\Psi[n]\rangle$ in Eq. (52) yields the charge density $n$ and minimizes $\langle \Psi|\hat{T}_{\mathrm{D}} + \hat{W}|\Psi\rangle$. The charge-conjugated state $\hat{C}|\Psi[n]\rangle$, where $\hat{C}$ is the charge-conjugation operator in Fock space (see Appendix A), yields the charge density $-n$ since

$$\langle \Psi[n]|\hat{C}^\dagger \hat{n}(\vec{r})\hat{C}|\Psi[n]\rangle = -\langle \Psi[n]|\hat{n}(\vec{r})|\Psi[n]\rangle = -n(\vec{r}), \tag{100}$$

where we have used the antisymmetry of the density operator under charge conjugation, $\hat{C}^\dagger \hat{n}(\vec{r})\hat{C} = -\hat{n}(\vec{r})$ [Eq. (144)]. Moreover, the charge-conjugated state $\hat{C}|\Psi[n]\rangle$ minimizes $\langle\Psi|\hat{T}_\text{D} + \hat{W}|\Psi\rangle$ since

$$\langle\Psi[n]|\hat{C}^\dagger(\hat{T}_\text{D} + \hat{W})\hat{C}|\Psi[n]\rangle = \langle\Psi[n]|\hat{T}_\text{D} + \hat{W}|\Psi[n]\rangle\,, \tag{101}$$

since both $\hat{T}_\text{D}$ and $\hat{W}$ are symmetric under charge conjugation [Eqs. (143) and (148)]. We thus conclude that

$$\hat{C}|\Psi[n]\rangle = |\Psi[-n]\rangle\,, \tag{102}$$

and that the universal density functional is symmetric under charge conjugation

$$F[n] = F[-n]\,. \tag{103}$$

Similarly, the KS determinant state in Eq. (55) transforms as

$$\hat{C}|\Phi[n]\rangle = |\Phi[-n]\rangle\,, \tag{104}$$

and the functionals $T_\text{s}[n]$, $E_\text{H}[n]$, $E_\text{x}[n]$, and $E_\text{c}[n]$ are all symmetric under charge conjugation

$$T_\text{s}[n] = T_\text{s}[-n]\,, \tag{105}$$

$$E_\text{H}[n] = E_\text{H}[-n]\,, \tag{106}$$

$$E_\text{x}[n] = E_\text{x}[-n]\,, \tag{107}$$

$$E_\text{c}[n] = E_\text{c}[-n]\,. \tag{108}$$

In other words, these functionals must be even functionals of the charge density. Consequently, their functional derivatives with respect to $n(\vec{r})$ must be odd functionals of the charge density. This is particularly obvious for the Coulomb contribution to the Hartree energy in Eq. (78).

**Uniform coordinate scaling relations**

In non-relativistic DFT, the uniform coordinate scaling relations [61–63] are important constraints on the density functionals. We show how to generalize them for the present RDFT.

Since there is generally no concept of wave function in the present relativistic theory, we cannot define coordinate scaling on wave functions, as normally done. Instead, we must work in Fock space and we thus define an unitary uniform coordinate scaling operator $\hat{S}_\gamma$ which transforms the Dirac field operator as

$$\hat{S}_\gamma^\dagger \hat{\psi}(\vec{r})\hat{S}_\gamma = \gamma^{3/2}\hat{\psi}(\gamma\vec{r})\,, \tag{109}$$

where $\gamma \in ]0, +\infty[$ is a scaling factor, and similarly for the separate electron and positron field operators in Eq. (137), i.e. $\hat{S}_\gamma^\dagger \hat{\psi}_+(\vec{r})\hat{S}_\gamma = \gamma^{3/2}\hat{\psi}_+(\gamma\vec{r})$ and $\hat{S}_\gamma^\dagger \hat{\psi}_-(\vec{r})\hat{S}_\gamma = \gamma^{3/2}\hat{\psi}_-(\gamma\vec{r})$. The one-particle density-matrix and density operators transform as

$$\hat{S}_\gamma^\dagger \hat{\mathbf{n}}_1(\vec{r}, \vec{r}\,')\, \hat{S}_\gamma = \gamma^3 \hat{\mathbf{n}}_1(\gamma\vec{r}, \gamma\vec{r}\,')\,, \tag{110}$$

and

$$\hat{S}_\gamma^\dagger \hat{n}(\vec{r})\, \hat{S}_\gamma = \gamma^3 \hat{n}(\gamma\vec{r})\,, \tag{111}$$

while the pair density-matrix operator transforms as

$$\hat{S}_\gamma^\dagger \, \hat{\mathbf{n}}_2(\vec{r}_1, \vec{r}_2) \, \hat{S}_\gamma = \gamma^6 \hat{\mathbf{n}}_2(\gamma \vec{r}_1, \gamma \vec{r}_2). \tag{112}$$

Since the scaling relations involve scaling the speed of light $c$, we will explicitly indicate in this section the dependence on $c$. A state $|\Psi^{\lambda,c}[n]\rangle$ in Eq. (82) for any coupling constant $\lambda$ and speed of light $c$ yields the charge density $n$ and minimizes $\langle \Psi | \hat{T}_D^c + \lambda \hat{W} | \Psi \rangle$. The scaled state

$$|\Psi_\gamma^{\lambda,c}[n]\rangle = \hat{S}_\gamma |\Psi^{\lambda,c}[n]\rangle, \tag{113}$$

yields the scaled charge density [see Eq. (111)]

$$n_\gamma(\vec{r}) = \gamma^3 n(\gamma \vec{r}), \tag{114}$$

and minimizes $\langle \Psi | \hat{T}_D^{c\gamma} + \lambda \gamma \hat{W} | \Psi \rangle$ since

$$\langle \Psi_\gamma^{\lambda,c}[n] | \hat{T}_D^{c\gamma} + \lambda \gamma \hat{W} | \Psi_\gamma^{\lambda,c}[n] \rangle = \gamma^2 \langle \Psi^{\lambda,c}[n] | \hat{T}_D^c + \lambda \hat{W} | \Psi^{\lambda,c}[n] \rangle, \tag{115}$$

where we have used Eqs. (110) and (112). We thus conclude that the scaled state $|\Psi_\gamma^{\lambda,c}[n]\rangle$ at coupling constant $\lambda$ and speed of light $c$ corresponds to the state at scaled density $n_\gamma$, scaled coupling constant $\lambda\gamma$, and scaled speed of light $c\gamma$

$$|\Psi_\gamma^{\lambda,c}[n]\rangle = |\Psi^{\lambda\gamma,c\gamma}[n_\gamma]\rangle, \tag{116}$$

or, equivalently,

$$|\Psi_\gamma^{\lambda/\gamma,c/\gamma}[n]\rangle = |\Psi^{\lambda,c}[n_\gamma]\rangle, \tag{117}$$

and that the universal density functional satisfies the scaling relation

$$F^{\lambda\gamma,c\gamma}[n_\gamma] = \gamma^2 F^{\lambda,c}[n], \tag{118}$$

or, equivalently,

$$F^{\lambda,c}[n_\gamma] = \gamma^2 F^{\lambda/\gamma,c/\gamma}[n]. \tag{119}$$

At $\lambda = 0$, we find the scaling relation of the KS single-determinant state

$$|\Phi_\gamma^{c/\gamma}[n]\rangle = |\Phi^c[n_\gamma]\rangle, \tag{120}$$

which directly leads to the scaling relation for the non-interacting kinetic density functional

$$T_s^c[n_\gamma] = \gamma^2 T_s^{c/\gamma}[n], \tag{121}$$

and for the Hartree and exchange density functionals

$$E_H^c[n_\gamma] = \gamma E_H^{c/\gamma}[n] \quad \text{and} \quad E_x^c[n_\gamma] = \gamma E_x^{c/\gamma}[n]. \tag{122}$$

The correlation density functional has the same scaling as $F^{\lambda,c}[n]$

$$E_c^{\lambda,c}[n_\gamma] = \gamma^2 E_c^{\lambda/\gamma,c/\gamma}[n], \tag{123}$$

and, in particular, for $\lambda = 1$

$$E_c^c[n_\gamma] = \gamma^2 E_c^{1/\gamma,c/\gamma}[n]. \tag{124}$$

These scaling relations imply that the low-density limit ($\gamma \to 0$) corresponds to the non-relativistic limit ($c \to \infty$), while the high-density limit ($\gamma \to \infty$) corresponds to the ultra-relativistic limit ($m \to 0$ where $m$ is the electron mass).

In the low-density limit, we indeed recover the well-known behaviors of the non-relativistic density functionals. After removing the rest-mass energy of $N$ electrons, $Nmc^2$, the non-interacting kinetic-energy functional scales quadratically as $\gamma \to 0$

$$T_s^c[n_\gamma] - Nmc^2 \underset{\gamma \to 0}{\sim} \gamma^2 T_s^{\text{NR}}[n], \tag{125}$$

where $T_s^{\text{NR}}[n] = \lim_{c \to \infty}(T_s^c[n] - Nmc^2)$ is the non-relativistic (NR) non-interacting kinetic-energy functional. The Hartree and exchange functionals scale linearly as $\gamma \to 0$

$$E_{\text{H}}^c[n_\gamma] \underset{\gamma \to 0}{\sim} \gamma E_{\text{H}}^{\text{NR}}[n] \quad \text{and} \quad E_{\text{x}}^c[n_\gamma] \underset{\gamma \to 0}{\sim} \gamma E_{\text{x}}^{\text{NR}}[n], \tag{126}$$

where $E_{\text{H}}^{\text{NR}}[n] = \lim_{c \to \infty} E_{\text{H}}^c[n] = E_{\text{H}}^{\text{C}}[n]$ [Eq. (78)] and $E_{\text{x}}^{\text{NR}}[n] = \lim_{c \to \infty} E_{\text{x}}^c[n]$ are the non-relativistic Hartree and exchange functionals. The correlation functional also scales linearly as $\gamma \to 0$

$$E_{\text{c}}^c[n_\gamma] \underset{\gamma \to 0}{\sim} \gamma W_{\text{c}}^{\text{NR,SCE}}[n], \tag{127}$$

where $W_{\text{c}}^{\text{NR,SCE}}[n] = \lim_{\lambda \to \infty} E_{\text{c}}^{\text{NR},\lambda}[n]/\lambda$ is the non-relativistic strictly-correlated-electron (SCE) correlation functional [64–67] obtained from the non-relativistic correlation functional along the adiabatic connection $E_{\text{c}}^{\text{NR},\lambda}[n] = \lim_{c \to \infty} E_{\text{c}}^{c,\lambda}[n]$ [see Eq. (84)] in the limit of infinite coupling constant $\lambda \to \infty$. The low-density limit is also called the strong-interaction limit since in this limit the Hartree, exchange, and correlation energies dominate over the non-interacting kinetic energy.

The high-density limit of the relativistic density functionals is more exotic. In this limit, the rest-mass term in the Dirac operator becomes negligible in comparison to the kinetic term, i.e. $\mathbf{D}^{c/\gamma}(\vec{r}) = (c/\gamma)(\vec{\alpha} \cdot \vec{p}) + \boldsymbol{\beta}\, mc^2/\gamma^2 \underset{\gamma \to \infty}{\sim} (c/\gamma)(\vec{\alpha} \cdot \vec{p})$, and consequently the non-interacting kinetic-energy functional scales linearly as $\gamma \to \infty$

$$T_s^c[n_\gamma] \underset{\gamma \to \infty}{\sim} \gamma T_s^{c,\text{UR}}[n], \tag{128}$$

where $T_s^{c,\text{UR}}[n] = \lim_{m \to 0} T_s^c[n]$ is the ultra-relativistic (UR) non-interacting kinetic-energy functional obtained by letting the electron mass going to zero in the Dirac operator. This is in contrast with the quadratic scaling of the non-relativistic kinetic-energy functional, i.e. $T_s^{\text{NR}}[n_\gamma] = \gamma^2 T_s^{\text{NR}}[n]$. The Hartree and exchange functionals also scale linearly as $\gamma \to \infty$

$$E_{\text{H}}^c[n_\gamma] \underset{\gamma \to \infty}{\sim} \gamma E_{\text{H}}^{c,\text{UR}}[n] \quad \text{and} \quad E_{\text{x}}^c[n_\gamma] \underset{\gamma \to \infty}{\sim} \gamma E_{\text{x}}^{c,\text{UR}}[n], \tag{129}$$

where $E_{\text{H}}^{c,\text{UR}}[n] = \lim_{m \to 0} E_{\text{H}}^c[n]$ and $E_{\text{x}}^{c,\text{UR}}[n] = \lim_{m \to 0} E_{\text{x}}^c[n]$ are the ultra-relativistic Hartree and exchange functionals. This is similar to the linear scaling of the non-relativistic Hartree and exchange functionals $E_{\text{H}}^{\text{NR}}[n_\gamma] = \gamma E_{\text{H}}^{\text{NR}}[n]$ and $E_{\text{x}}^{\text{NR}}[n_\gamma] = \gamma E_{\text{x}}^{\text{NR}}[n]$. Finally, the correlation functional scales linearly as $\gamma \to \infty$

$$E_{\text{c}}^c[n_\gamma] \underset{\gamma \to \infty}{\sim} \gamma E_{\text{c}}^{c,\text{UR}}[n], \tag{130}$$

where $E_{\text{c}}^{c,\text{UR}}[n] = \lim_{m \to 0} E_{\text{c}}^c[n]$ is the ultra-relativistic correlation functional. This is again in contrast with the non-relativistic case where the correlation functional goes to a constant as $\gamma \to \infty$, for a KS Hamiltonian with a non-degenerate ground state,

$\lim_{\gamma \to \infty} E_c^{\text{NR}}[n_\gamma] = E_c^{\text{NR,GL2}}[n]$, where $E_c^{\text{NR,GL2}}[n]$ is the second-order Görling-Levy (GL2) correlation energy [68, 69]. Hence, in the relativistic case, the high-density limit is no longer a weak-interaction or weak-correlation limit since $T_s^c[n_\gamma]$, $E_H^c[n_\gamma]$, $E_x^c[n_\gamma]$, and $E_c^c[n_\gamma]$ all scale linearly in $\gamma$. In particular, the divergence of the relativistic correlation functional in the high-density limit has important implications for relativistic functional development. Indeed, many non-relativistic correlation functionals, such as the Perdew-Burke-Ernzerhof (PBE) one [70], have been designed to saturate in the high-density limit. Hence, these non-relativistic correlation functionals should be rethought so as to satisfy Eq. (130).

The same scaling relations apply in the no-pair approximation, as well as in the npvp variant of Eq. (91). In the configuration-space approach of the no-pair approximation, these scaling relations could be obtained using the minmax principle (see Ref. [52]).

In the non-relativistic theory, the high-density limit is realized in atomic ions in the limit of large nuclear charge, $Z \to \infty$, at fixed electron number $N$ (see Refs. [71, 72]). In a relativistic setting, the relation between the high-density limit and the large nuclear-charge limit is more complicated due to the scaling of the speed of light [50]. However, we note that numerical studies show that relativistic no-pair and beyond-no-pair correlation energies (calculated with respect to HF) of two-electron atoms diverge as $Z$ increases [50, 73], which is in line with the divergence of $E_c^c[n_\gamma]$ as $\gamma \to \infty$ [Eq. (130)].

Finally, for $\gamma = \lambda$, the scaling relation in Eq. (123) gives an expression for the correlation functional along the adiabatic connection at coupling constant $\lambda$

$$E_c^{\lambda,c}[n] = \lambda^2 E_c^{c/\lambda}[n_{1/\lambda}], \tag{131}$$

which could be useful for analyzing approximate correlation functionals and for developing a relativistic extension of the multideterminant KS scheme of Refs. [74, 75].

## 3.5 Local-density approximation

The LDA is usually the first approximation considered in DFT. In the present relativistic theory, the LDA exchange-correlation functional may be written as

$$E_{\text{xc}}^{\text{LDA}}[n] = \int |n(\vec{r})| \epsilon_{\text{xc}}^{\text{RHEG}}(|n(\vec{r})|) \mathrm{d}\vec{r}, \tag{132}$$

where $\epsilon_{\text{xc}}^{\text{RHEG}}(n)$ is the exchange-correlation energy per particle of the relativistic homogeneous electron gas (RHEG) of constant charge density $n \in [0, +\infty[$. To deal with the possibility of having negative charge densities $n(\vec{r})$ at some points of space in the inhomogeneous system [see discussion in the paragraph after Eq. (55)], we have used the absolute value of the charge density. On the one hand, this permits to satisfy charge-conjugation symmetry [Eqs. (107) and (108)], but, on the other hand, it introduces discontinuities in the corresponding potential at the points of space where $n(\vec{r})$ changes sign. Whether using the absolute value of the charge density is the right thing to do is thus unsure and should be further studied.

Since the RHEG has a spatially constant charge density, its KS potential $v + v_{\text{Hxc}}$ in Eq. (61) must necessarily be a spatial constant as well. Since the KS potential does not depend on spinor indices either (contrary to the HF potential), the KS orbitals of the RHEG are thus simply the eigenfunctions of the free Dirac equation. In other words, due to translational symmetry, the KS vacuum state $|\tilde{0}\rangle$ of the RHEG is equal to the free vacuum state $|0\rangle$. Consequently, the vacuum-polarization one-particle density matrix in Eq. (60) vanishes for the RHEG and the LDA exchange functional does not contain any vacuum-polarization contribution, i.e. $E_x^{\text{LDA}}[n] = \tilde{E}_x^{\text{LDA}}[n]$ [Eq. (75)] or $\tilde{E}_x^{\text{vp,LDA}}[n] = 0$ [Eq. (76)]. Similarly, for the LDA correlation functional, we have $E_c^{\text{LDA}}[n] = \tilde{E}_c^{\text{LDA}}[n]$ [Eq. (88)] or $\tilde{E}_c^{\text{vp,LDA}}[n] = 0$ [Eq. (89)], but $E_c^{\text{LDA}}[n]$

still contains vacuum contributions via the correlation pair-density matrix $\tilde{\mathbf{n}}_{2,c}^{\lambda}(\vec{r}_1, \vec{r}_2)$ of the RHEG.

Moreover, for the same reason, the KS orbitals of the RHEG obtained in the no-pair approximation [Eq. (95)] are also necessarily the eigenfunctions of the free Dirac equation, and thus the no-pair approximation has no impact on the LDA exchange functional, i.e. $E_x^{\text{LDA}}[n] = E_x^{\text{np,LDA}}[n]$. By contrast, the no-pair approximation or its npvp variant [Eq. (92)] do have an impact of the LDA correlation functional, i.e. $E_c^{\text{LDA}}[n] \neq E_c^{\text{npvp,LDA}}[n] = E_c^{\text{np,LDA}}[n]$, since the vacuum contributions are now suppressed from $\tilde{\mathbf{n}}_{2,c}^{\lambda}(\vec{r}_1, \vec{r}_2)$.

The exchange energy per particle of the RHEG for the Coulomb interaction of Eq. (15) is [4,76] (see, also, Ref. [51])

$$
\begin{aligned}
\epsilon_x^{\text{RHEG,C}}(n) = & -\frac{3\,k_F}{4\pi}\Bigg[\frac{5}{6} + \frac{1}{3}\tilde{c}^2 + \frac{2}{3}\sqrt{1+\tilde{c}^2}\,\text{arcsinh}\left(\frac{1}{\tilde{c}}\right) - \frac{1}{3}\left(1+\tilde{c}^2\right)^2 \ln\left(1+\frac{1}{\tilde{c}^2}\right) \\
& -\frac{1}{2}\left(\sqrt{1+\tilde{c}^2} - \tilde{c}^2\text{arcsinh}\left(\frac{1}{\tilde{c}}\right)\right)^2\Bigg],
\end{aligned}
\tag{133}
$$

where $k_F = (3\pi^2 n)^{1/3}$ is the Fermi wave vector and $\tilde{c} = mc/k_F$ is a relativistic parameter. The exchange energy per particle for the Breit interaction of Eq. (16) has a similar form [77] (see, also, Ref. [51])

$$
\begin{aligned}
\epsilon_x^{\text{RHEG,B}}(n) = & \frac{3\,k_F}{4\pi}\Bigg[1 - 2\left(1+\tilde{c}^2\right)\left(1 - \tilde{c}^2\left(-2\ln(\tilde{c}) + \ln\left(1+\tilde{c}^2\right)\right)\right) \\
& +2\left(\sqrt{1+\tilde{c}^2} - \tilde{c}^2\text{arcsinh}\left(\frac{1}{\tilde{c}}\right)\right)^2\Bigg].
\end{aligned}
\tag{134}
$$

Note that these expressions are valid for an arbitrary speed of light $c$. The dependence on $c$ via the adimensional parameter $\tilde{c}$ is necessary for the LDA exchange functional to satisfy the scaling relation of Eq. (122). Note that the Breit exchange energy per particle is an approximation to the exchange energy per particle obtained with the transverse component of the full QED photon propagator [3,4,76]. The exchange energy per particle obtained with the full QED photon propagator has in fact a simpler expression than the Coulomb-Breit one, thanks to the cancellation of many terms between the Coulomb and transverse components,

$$
\epsilon_x^{\text{QED}}(n) = -\frac{3\,k_F}{4\pi}\Bigg[1 - \frac{3}{2}\left(\sqrt{1+\tilde{c}^2} - \tilde{c}^2\text{arcsinh}\left(\frac{1}{\tilde{c}}\right)\right)^2\Bigg].
\tag{135}
$$

The Coulomb-Breit exchange energy per particle is a good approximation to the exchange energy per particle obtained with the full QED photon propagator for $k_F \lesssim c$ [51]. In any case, the LDA exchange functional corresponding to the present RDFT is given by Eqs. (133) and (134), and not by Eq. (135).

Contrary to the case of exchange, the correlation energy per particle of the RHEG cannot be calculated analytically. It has been estimated numerically at the level of the relativistic random-phase approximation, using either the no-sea approximation (which includes parts of the vacuum contributions) or the no-pair approximation, and the full QED photon propagator or the Coulomb-Breit interaction [78,79] (see also Refs. [7–9, 14, 80–82]). However, to the best of our knowledge, these calculations were done for the fixed physical value of the speed of light. Therefore, we do not have the dependence on $c$ and we cannot apply the scaling relation of Eq. (124) or (131). More work seems necessary to construct the LDA correlation functional including the dependence on $c$ with or without the no-pair approximation.

## 4 Conclusions

In this work, we have examined a RDFT based on an effective QED without the photon degrees of freedom. The formalism is appealing since it is simpler than RDFT based on full QED. We have used this formalism to unambiguously define density functionals in the no-pair approximation, thus making a closer contact with calculations done in practice, and to study some exact properties of the involved functionals, namely charge-conjugation symmetry and uniform coordinate scaling. The formalism has also the advantage to be easily extended to multideterminant KS schemes which combine wave-function methods with density functionals based on a decomposition the electron-electron interaction (see, e.g., Refs. [74, 83, 84]).

In possible future works on the present RDFT, one may study whether this approach can be made mathematically rigorous, one may develop density-functional approximations for this approach, one may examine the extension to functionals of the charge current density or of the one-particle density matrix, and one may implement this approach for example for calculations of vacuum-polarization effects in heavy atoms. This last goal would require the development of practical regularization/renormalization procedures.

## Acknowledgements

I thank Christian Brouder, Emmanuel Giner, Mathieu Lewin, Julien Paquier, and Trond Saue for discussions and/or comments on the manuscript.

## A  Charge-conjugation symmetry of the electron-positron Hamiltonian

Under charge conjugation, the Dirac field operator transforms as (see, e.g., Refs. [9,43,45,85])

$$\hat{C}\hat{\psi}(\vec{r})\hat{C}^{\dagger} = \mathbf{C}\hat{\psi}^{\dagger\mathrm{T}}(\vec{r}), \qquad (136)$$

with the unitary charge-conjugation symmetry operator in Fock space $\hat{C}$, the unitary matrix $\mathbf{C} = -i\boldsymbol{\alpha}_y\boldsymbol{\beta}$ defined up to an unimportant phase factor, and $^{\mathrm{T}}$ designating the matrix transposition. If we decompose the Dirac field operator into free electron and positron field contributions

$$\hat{\psi}(\vec{r}) = \hat{\psi}_+(\vec{r}) + \hat{\psi}_-(\vec{r}), \qquad (137)$$

with $\hat{\psi}_+(\vec{r}) = \sum_{p\in\mathrm{PS}}\hat{b}_p\psi_p(\vec{r})$ and $\hat{\psi}_-(\vec{r}) = \sum_{p\in\mathrm{NS}}\hat{d}_p^{\dagger}\psi_p(\vec{r})$ in which $\{\psi_p\}$ is the set of eigenfunctions of the free Dirac equation, then charge conjugation interchanges these contributions as

$$\hat{C}\hat{\psi}_+(\vec{r})\hat{C}^{\dagger} = \mathbf{C}\hat{\psi}_-^{\dagger\mathrm{T}}(\vec{r}), \qquad (138)$$

$$\hat{C}\hat{\psi}_-(\vec{r})\hat{C}^{\dagger} = \mathbf{C}\hat{\psi}_+^{\dagger\mathrm{T}}(\vec{r}), \qquad (139)$$

or, writing explicitly the spinor components, $\hat{C}\hat{\psi}_{+,\sigma}(\vec{r})\hat{C}^{\dagger} = \sum_{\sigma'} C_{\sigma\sigma'}\hat{\psi}_{-,\sigma'}^{\dagger}(\vec{r})$ and $\hat{C}\hat{\psi}_{-,\sigma}(\vec{r})\hat{C}^{\dagger} = \sum_{\sigma'} C_{\sigma\sigma'}\hat{\psi}_{+,\sigma'}^{\dagger}(\vec{r})$. Let us stress that Eqs. (138) and (139) are only valid when using the orbitals of the free Dirac equation $\{\psi_p\}$ and not arbitrary orbitals $\{\tilde{\psi}_p\}$. These equations allow us to find the transformation under charge conjugation of the electron-positron Hamiltonian in Eq. (7) expressed with normal ordering with respect to the free vacuum state.

In terms of the free electron and positron field operators, the one-particle density-matrix operator in Eq. (11) has the expression

$$\hat{n}_{1,\rho\sigma}(\vec{r},\vec{r}\,') = \hat{\psi}_{+,\sigma}^{\dagger}(\vec{r}\,')\hat{\psi}_{+,\rho}(\vec{r}) + \hat{\psi}_{+,\sigma}^{\dagger}(\vec{r}\,')\hat{\psi}_{-,\rho}(\vec{r}) + \hat{\psi}_{-,\sigma}^{\dagger}(\vec{r}\,')\hat{\psi}_{+,\rho}(\vec{r}) - \hat{\psi}_{-,\rho}(\vec{r})\hat{\psi}_{-,\sigma}^{\dagger}(\vec{r}\,'),$$
(140)

which becomes under charge conjugation

$$
\begin{aligned}
\hat{C}\hat{n}_{1,\rho\sigma}(\vec{r},\vec{r}\,')\hat{C}^{\dagger} &= \sum_{\rho'\sigma'} C_{\rho\rho'}[\hat{\psi}_{-,\sigma'}(\vec{r}\,')\hat{\psi}_{-,\rho'}^{\dagger}(\vec{r}) + \hat{\psi}_{-,\sigma'}(\vec{r}\,')\hat{\psi}_{+,\rho'}^{\dagger}(\vec{r}) \\
&\quad + \hat{\psi}_{+,\sigma'}(\vec{r}\,')\hat{\psi}_{-,\rho'}^{\dagger}(\vec{r}) - \hat{\psi}_{+,\rho'}^{\dagger}(\vec{r})\hat{\psi}_{+,\sigma'}(\vec{r}\,')]C_{\sigma'\sigma}^{\dagger} \\
&= -\sum_{\rho'\sigma'} C_{\rho\rho'}\hat{n}_{1,\sigma'\rho'}(\vec{r}\,',\vec{r})C_{\sigma'\sigma}^{\dagger},
\end{aligned}
$$
(141)

or, in matrix form,

$$\hat{C}\hat{\mathbf{n}}_1(\vec{r},\vec{r}\,')\hat{C}^{\dagger} = -\mathbf{C}\hat{\mathbf{n}}_1^{\mathrm{T}}(\vec{r}\,',\vec{r})\mathbf{C}^{\dagger}.$$
(142)

From this, we deduce that the Dirac kinetic + rest mass operator $\hat{T}_{\mathrm{D}}$ in Eq. (8) is symmetric under charge conjugation

$$
\begin{aligned}
\hat{C}\hat{T}_{\mathrm{D}}\hat{C}^{\dagger} &= -\int \mathrm{Tr}[\mathbf{D}(\vec{r})\mathbf{C}\hat{\mathbf{n}}_1^{\mathrm{T}}(\vec{r}\,',\vec{r})\mathbf{C}^{\dagger}]_{\vec{r}\,'=\vec{r}}\,\mathrm{d}\vec{r} \\
&= -\int \mathrm{Tr}[\mathbf{C}^{\dagger}\mathbf{D}(\vec{r})\mathbf{C}\hat{\mathbf{n}}_1^{\mathrm{T}}(\vec{r}\,',\vec{r})]_{\vec{r}\,'=\vec{r}}\,\mathrm{d}\vec{r} \\
&= \int \mathrm{Tr}[\mathbf{D}(\vec{r})\hat{\mathbf{n}}_1(\vec{r},\vec{r}\,')]_{\vec{r}\,'=\vec{r}}\,\mathrm{d}\vec{r} \\
&= \hat{T}_{\mathrm{D}},
\end{aligned}
$$
(143)

where we have used $\mathbf{C}^{\dagger}\mathbf{D}(\vec{r})\mathbf{C} = -\mathbf{D}^*(\vec{r}) = -c\,(\vec{\boldsymbol{\alpha}}^* \cdot \vec{p}\,^*) - \boldsymbol{\beta}\,mc^2$ and the third equality in Eq. (143) comes from the hermiticity of $\vec{\boldsymbol{\alpha}}$, i.e. $\vec{\boldsymbol{\alpha}}^* = \vec{\boldsymbol{\alpha}}^{\mathrm{T}}$, and the self-adjointness of $\vec{p}$. Moreover, from Eq. (142), we find the expected antisymmetry of the opposite charge density operator under charge conjugation

$$\hat{C}\hat{n}(\vec{r})\hat{C}^{\dagger} = -\hat{n}(\vec{r}),$$
(144)

which immediately shows that the external potential operator $\hat{V}$ in Eq. (10) is also antisymmetric

$$\hat{C}\hat{V}\hat{C}^{\dagger} = -\hat{V}.$$
(145)

A similar calculation gives the transformation of the pair density-matrix operator in Eq. (12) under charge conjugation

$$\hat{C}\hat{n}_{2,\rho\upsilon\sigma\tau}(\vec{r}_1,\vec{r}_2)\hat{C}^{\dagger} = \sum_{\rho'\upsilon'\tau'\sigma'} C_{\rho\rho'}C_{\upsilon\upsilon'}\hat{n}_{2,\tau'\sigma'\upsilon'\rho'}(\vec{r}_2,\vec{r}_1)C_{\tau'\tau}^{\dagger}C_{\sigma'\sigma}^{\dagger},$$
(146)

or, in matrix notation,

$$\hat{C}\hat{\mathbf{n}}_2(\vec{r}_1,\vec{r}_2)\hat{C}^{\dagger} = (\mathbf{C}\otimes\mathbf{C})\hat{\mathbf{n}}_2^{\mathrm{T}}(\vec{r}_2,\vec{r}_1)(\mathbf{C}\otimes\mathbf{C})^{\dagger},$$
(147)

where $\otimes$ is the matrix tensor product. This shows that the two-particle interaction operator $\hat{W}$ in Eq. (10) is symmetric under charge conjugation

$$
\begin{aligned}
\hat{C}\hat{W}\hat{C}^\dagger &= \frac{1}{2}\iint \mathrm{Tr}[\mathbf{w}(\vec{r}_1,\vec{r}_2)(\mathbf{C}\otimes\mathbf{C})\hat{\mathbf{n}}_2^{\mathrm{T}}(\vec{r}_2,\vec{r}_1)(\mathbf{C}\otimes\mathbf{C})^\dagger]\mathrm{d}\vec{r}_1\mathrm{d}\vec{r}_2 \\
&= \frac{1}{2}\iint \mathrm{Tr}[(\mathbf{C}\otimes\mathbf{C})^\dagger\mathbf{w}(\vec{r}_1,\vec{r}_2)(\mathbf{C}\otimes\mathbf{C})\hat{\mathbf{n}}_2^{\mathrm{T}}(\vec{r}_2,\vec{r}_1)]\mathrm{d}\vec{r}_1\mathrm{d}\vec{r}_2 \\
&= \frac{1}{2}\iint \mathrm{Tr}[\mathbf{w}(\vec{r}_1,\vec{r}_2)\hat{\mathbf{n}}_2(\vec{r}_2,\vec{r}_1)]\mathrm{d}\vec{r}_1\mathrm{d}\vec{r}_2 \\
&= \hat{W},
\end{aligned}
\tag{148}
$$

where we have used $(\mathbf{C}\otimes\mathbf{C})^\dagger\mathbf{w}(\vec{r}_1,\vec{r}_2)(\mathbf{C}\otimes\mathbf{C}) = \mathbf{w}(\vec{r}_1,\vec{r}_2) = \mathbf{w}^{\mathrm{T}}(\vec{r}_1,\vec{r}_2)$ and $\mathbf{w}(\vec{r}_1,\vec{r}_2) = \mathbf{w}(\vec{r}_2,\vec{r}_1)$.

In conclusion, we thus have found the expected transformation of the electron-positron Hamiltonian under charge conjugation

$$
\hat{C}\hat{H}[\nu]\hat{C}^\dagger = \hat{H}[-\nu].
\tag{149}
$$

## B  Alternative definition of the electron-positron Hamiltonian

As an alternative to the definition of the electron-positron Hamiltonian based on normal ordering with respect to the free vacuum state in Eq. (7), an electron-positron Hamiltonian based on commutators and anticommutators (which we indicate by using the superscript c) of Dirac field operators can be defined as

$$
\hat{H}^{\mathrm{c}} = \hat{T}_{\mathrm{D}}^{\mathrm{c}} + \hat{W}^{\mathrm{c}} + \hat{V}^{\mathrm{c}},
\tag{150}
$$

with

$$
\hat{T}_{\mathrm{D}}^{\mathrm{c}} = \int \mathrm{Tr}[\mathbf{D}(\vec{r})\hat{\mathbf{n}}_1^{\mathrm{c}}(\vec{r},\vec{r}\,')]_{\vec{r}\,'=\vec{r}}\,\mathrm{d}\vec{r},
\tag{151}
$$

and

$$
\hat{W}^{\mathrm{c}} = \frac{1}{2}\iint \mathrm{Tr}[\mathbf{w}(\vec{r}_1,\vec{r}_2)\hat{\mathbf{n}}_2^{\mathrm{c}}(\vec{r}_1,\vec{r}_2)]\mathrm{d}\vec{r}_1\mathrm{d}\vec{r}_2,
\tag{152}
$$

and

$$
\hat{V}^{\mathrm{c}} = \int \nu(\vec{r})\hat{n}^{\mathrm{c}}(\vec{r})\,\mathrm{d}\vec{r}.
\tag{153}
$$

In these expressions, $\hat{\mathbf{n}}_1^{\mathrm{c}}(\vec{r},\vec{r}\,')$ is an one-particle density matrix operator defined as a commutator of Dirac field operators

$$
\hat{n}_{1,\rho\sigma}^{\mathrm{c}}(\vec{r},\vec{r}\,') = \frac{1}{2}\left[\hat{\psi}_\sigma^\dagger(\vec{r}\,'),\hat{\psi}_\rho(\vec{r})\right],
\tag{154}
$$

$\hat{n}^{\mathrm{c}}(\vec{r}) = \mathrm{Tr}[\hat{\mathbf{n}}_1^{\mathrm{c}}(\vec{r},\vec{r})]$ is the associated opposite charge density operator, and similarly $\hat{\mathbf{n}}_2^{\mathrm{c}}(\vec{r}_1,\vec{r}_2)$ is a pair density-matrix operator defined as an anticommutator of products of Dirac field operators

$$
\hat{n}_{2,\rho\upsilon\sigma\tau}^{\mathrm{c}}(\vec{r}_1,\vec{r}_2) = \frac{1}{2}\left\{\hat{\psi}_\tau^\dagger(\vec{r}_2)\hat{\psi}_\sigma^\dagger(\vec{r}_1),\hat{\psi}_\rho(\vec{r}_1)\hat{\psi}_\upsilon(\vec{r}_2)\right\}.
\tag{155}
$$

Whereas the commutator form in Eq. (154) is well known in the literature (see, e.g., Refs. [9,25]), the anticommutator form in Eq. (155) is, to the best of our knowledge, original to the present work. The commutator and the anticommutator in these definitions impose the correct transformation under charge conjugation without having to use normal ordering with respect to the free vacuum state. Indeed, using Eq. (136), it is straightforward to see that $\hat{\mathbf{n}}_1^c(\vec{r}, \vec{r}\,')$ correctly transforms as in Eq. (142)

$$\hat{C}\hat{\mathbf{n}}_1^c(\vec{r}, \vec{r}\,')\hat{C}^\dagger = -\mathbf{C}\hat{\mathbf{n}}_1^{cT}(\vec{r}\,', \vec{r})\mathbf{C}^\dagger, \tag{156}$$

and, similarly, $\hat{\mathbf{n}}_2^c(\vec{r}_1, \vec{r}_2)$ correctly transforms as in Eq. (147)

$$\hat{C}\hat{\mathbf{n}}_2^c(\vec{r}_1, \vec{r}_2)\hat{C}^\dagger = (\mathbf{C}\otimes\mathbf{C})\hat{\mathbf{n}}_2^{cT}(\vec{r}_2, \vec{r}_1)(\mathbf{C}\otimes\mathbf{C})^\dagger. \tag{157}$$

Using Wick's theorem, we can express $\hat{\mathbf{n}}_1^c(\vec{r}, \vec{r}\,')$ in terms of the one-particle density-matrix operator $\hat{\tilde{\mathbf{n}}}_1(\vec{r}, \vec{r}\,')$ defined with normal ordering with respect to the alternative no-particle vacuum state $|\tilde{0}\rangle$ in Eq. (27)

$$\hat{n}_{1,\rho\sigma}^c(\vec{r}, \vec{r}\,') = \hat{\tilde{n}}_{1,\rho\sigma}(\vec{r}, \vec{r}\,') + \tilde{n}_{1,\rho\sigma}^{c,vp}(\vec{r}, \vec{r}\,'), \tag{158}$$

with the associated vacuum-polarization one-particle density matrix

$$
\begin{aligned}
\tilde{n}_{1,\rho\sigma}^{c,vp}(\vec{r}, \vec{r}\,') &= \langle\tilde{0}|\hat{n}_{1,\rho\sigma}^c(\vec{r}, \vec{r}\,')|\tilde{0}\rangle \\
&= \frac{1}{2}\Big(\langle\tilde{0}|\hat{\psi}_\sigma^\dagger(\vec{r}\,')\hat{\psi}_\rho(\vec{r})|\tilde{0}\rangle - \langle\tilde{0}|\hat{\psi}_\rho(\vec{r})\hat{\psi}_\sigma^\dagger(\vec{r}\,')|\tilde{0}\rangle\Big) \\
&= \frac{1}{2}\Bigg(\sum_{p\in\text{NS}}\tilde{\psi}_{p,\sigma}^*(\vec{r}\,')\tilde{\psi}_{p,\rho}(\vec{r}) - \sum_{p\in\text{PS}}\tilde{\psi}_{p,\sigma}^*(\vec{r}\,')\tilde{\psi}_{p,\rho}(\vec{r})\Bigg).
\end{aligned}
\tag{159}
$$

Similarly, we can express $\hat{\mathbf{n}}_2^c(\vec{r}_1, \vec{r}_2)$ in terms of the pair density-matrix operator $\hat{\tilde{\mathbf{n}}}_2(\vec{r}_1, \vec{r}_2)$ defined with normal ordering with respect to the vacuum state $|\tilde{0}\rangle$ in Eq. (28)

$$
\begin{aligned}
\hat{n}_{2,\rho v\sigma\tau}^c(\vec{r}_1, \vec{r}_2) &= \hat{\tilde{n}}_{2,\rho v\sigma\tau}(\vec{r}_1, \vec{r}_2) + \tilde{n}_{1,v\tau}^{c,vp}(\vec{r}_2, \vec{r}_2)\hat{\tilde{n}}_{1,\rho\sigma}(\vec{r}_1, \vec{r}_1) + \tilde{n}_{1,\rho\sigma}^{c,vp}(\vec{r}_1, \vec{r}_1)\hat{\tilde{n}}_{1,v\tau}(\vec{r}_2, \vec{r}_2) \\
&\quad - \tilde{n}_{1,v\sigma}^{c,vp}(\vec{r}_2, \vec{r}_1)\hat{\tilde{n}}_{1,\rho\tau}(\vec{r}_1, \vec{r}_2) - \tilde{n}_{1,\rho\tau}^{c,vp}(\vec{r}_1, \vec{r}_2)\hat{\tilde{n}}_{1,v\sigma}(\vec{r}_2, \vec{r}_1) + \tilde{n}_{2,\rho v\sigma\tau}^{c,vp}(\vec{r}_1, \vec{r}_2),
\end{aligned}
\tag{160}
$$

with the associated vacuum-polarization pair density matrix

$$
\begin{aligned}
\tilde{n}_{2,\rho v\sigma\tau}^{c,vp}&(\vec{r}_1, \vec{r}_2) = \langle\tilde{0}|\hat{n}_{2,\rho v\sigma\tau}^c(\vec{r}_1, \vec{r}_2)|\tilde{0}\rangle \\
&= \frac{1}{2}\Big(\langle\tilde{0}|\hat{\psi}_\tau^\dagger(\vec{r}_2)\hat{\psi}_v(\vec{r}_2)|\tilde{0}\rangle\langle\tilde{0}|\hat{\psi}_\sigma^\dagger(\vec{r}_1)\hat{\psi}_\rho(\vec{r}_1)|\tilde{0}\rangle - \langle\tilde{0}|\hat{\psi}_\tau^\dagger(\vec{r}_2)\hat{\psi}_\rho(\vec{r}_1)|\tilde{0}\rangle\langle\tilde{0}|\hat{\psi}_\sigma^\dagger(\vec{r}_1)\hat{\psi}_v(\vec{r}_2)|\tilde{0}\rangle \\
&\quad + \langle\tilde{0}|\hat{\psi}_v(\vec{r}_2)\hat{\psi}_\tau^\dagger(\vec{r}_2)|\tilde{0}\rangle\langle\tilde{0}|\hat{\psi}_\rho(\vec{r}_1)\hat{\psi}_\sigma^\dagger(\vec{r}_1)|\tilde{0}\rangle - \langle\tilde{0}|\hat{\psi}_\rho(\vec{r}_1)\hat{\psi}_\tau^\dagger(\vec{r}_2)|\tilde{0}\rangle\langle\tilde{0}|\hat{\psi}_v(\vec{r}_2)\hat{\psi}_\sigma^\dagger(\vec{r}_1)|\tilde{0}\rangle\Big) \\
&= \frac{1}{2}\Bigg(\sum_{p,q\in\text{NS}}\tilde{\psi}_{p,\tau}^*(\vec{r}_2)\tilde{\psi}_{p,v}(\vec{r}_2)\tilde{\psi}_{q,\sigma}^*(\vec{r}_1)\tilde{\psi}_{q,\rho}(\vec{r}_1) - \sum_{p,q\in\text{NS}}\tilde{\psi}_{p,\tau}^*(\vec{r}_2)\tilde{\psi}_{p,\rho}(\vec{r}_1)\tilde{\psi}_{q,\sigma}^*(\vec{r}_1)\tilde{\psi}_{q,v}(\vec{r}_2) \\
&\quad + \sum_{p,q\in\text{PS}}\tilde{\psi}_{p,\tau}^*(\vec{r}_2)\tilde{\psi}_{p,v}(\vec{r}_2)\tilde{\psi}_{q,\sigma}^*(\vec{r}_1)\tilde{\psi}_{q,\rho}(\vec{r}_1) - \sum_{p,q\in\text{PS}}\tilde{\psi}_{p,\tau}^*(\vec{r}_2)\tilde{\psi}_{p,\rho}(\vec{r}_1)\tilde{\psi}_{q,\sigma}^*(\vec{r}_1)\tilde{\psi}_{q,v}(\vec{r}_2)\Bigg).
\end{aligned}
\tag{161}
$$

Similarly to what was done in Eq. (33), the electron-positron Hamiltonian in Eq. (150) can then be rewritten as

$$\hat{H}^c = \hat{\tilde{T}}_D + \hat{\tilde{W}} + \hat{\tilde{V}} + \hat{\tilde{V}}^{vp} + \tilde{E}_0^c, \tag{162}$$

where $\hat{\tilde{T}}_{\mathrm{D}}$, $\hat{\tilde{W}}$, and $\hat{\tilde{V}}$ have been already defined in Eqs. (34)-(36), and $\hat{\tilde{V}}^{\mathrm{vp}}$ and $\tilde{E}_0^{\mathrm{c}}$ are the vacuum-polarization potential and no-particle vacuum energy associated with this Hamiltonian. Similarly to Eq. (38), the vacuum-polarization potential can be written as

$$\hat{\tilde{V}}^{\mathrm{vp}} = \hat{\tilde{V}}_{\mathrm{d}}^{\mathrm{vp}} + \hat{\tilde{V}}_{\mathrm{x}}^{\mathrm{vp}}, \tag{163}$$

with a direct contribution

$$\hat{\tilde{V}}_{\mathrm{d}}^{\mathrm{vp}} = \int \mathrm{Tr}[\tilde{\mathbf{v}}_{\mathrm{d}}^{\mathrm{c,vp}}(\vec{r}_1)\hat{\tilde{\mathbf{n}}}(\vec{r}_1)]\mathrm{d}\vec{r}_1, \tag{164}$$

where $\tilde{v}_{\mathrm{d},\sigma\rho}^{\mathrm{c,vp}}(\vec{r}_1) = \sum_{\tau v}\int w_{\sigma\tau\rho v}(\vec{r}_1,\vec{r}_2)\tilde{n}_{v\tau}^{\mathrm{c,vp}}(\vec{r}_2)\mathrm{d}\vec{r}_2$ and $\tilde{n}_{v\tau}^{\mathrm{c,vp}}(\vec{r}_2) = \tilde{n}_{1,v\tau}^{\mathrm{c,vp}}(\vec{r}_2,\vec{r}_2)$, and an exchange contribution

$$\hat{\tilde{V}}_{\mathrm{x}}^{\mathrm{vp}} = \iint \mathrm{Tr}[\tilde{\mathbf{v}}_{\mathrm{x}}^{\mathrm{c,vp}}(\vec{r}_1,\vec{r}_2)\hat{\tilde{\mathbf{n}}}_1(\vec{r}_1,\vec{r}_2)]\mathrm{d}\vec{r}_1\mathrm{d}\vec{r}_2, \tag{165}$$

where $\tilde{v}_{\mathrm{x},\tau\rho}^{\mathrm{c,vp}}(\vec{r}_1,\vec{r}_2) = -\sum_{\sigma v}w_{\sigma\tau\rho v}(\vec{r}_1,\vec{r}_2)\tilde{n}_{1,v\sigma}^{\mathrm{c,vp}}(\vec{r}_2,\vec{r}_1)$. Finally, the associated no-particle vacuum energy can be written as

$$\begin{aligned}
\tilde{E}_0^{\mathrm{c}} &= \langle\tilde{0}|\hat{H}^{\mathrm{c}}|\tilde{0}\rangle \\
&= \int \mathrm{Tr}[\mathbf{D}(\vec{r})\tilde{\mathbf{n}}_1^{\mathrm{c,vp}}(\vec{r},\vec{r}\,')]_{\vec{r}\,'=\vec{r}}\,\mathrm{d}\vec{r} + \int v(\vec{r})\tilde{n}^{\mathrm{c,vp}}(\vec{r})\,\mathrm{d}\vec{r} \\
&\quad + \frac{1}{2}\iint \mathrm{Tr}[\mathbf{w}(\vec{r}_1,\vec{r}_2)\tilde{\mathbf{n}}_2^{\mathrm{c,vp}}(\vec{r}_1,\vec{r}_2)]\mathrm{d}\vec{r}_1\mathrm{d}\vec{r}_2.
\end{aligned} \tag{166}$$

As suggested by the fact that we used the same notation, it turns out that both the direct and exchange contributions to the vacuum-polarization potential in Eq. (163) are identical to the ones introduced in Eq. (38). This can be shown as follows. First, using the fact that the orbital rotation in Eq. (24) leaves invariant the following sum over orbitals

$$\sum_{p\in\mathrm{PS}}\tilde{\psi}_{p,\sigma}^*(\vec{r}\,')\tilde{\psi}_{p,\rho}(\vec{r}) + \sum_{p\in\mathrm{NS}}\tilde{\psi}_{p,\sigma}^*(\vec{r}\,')\tilde{\psi}_{p,\rho}(\vec{r}) = \sum_{p\in\mathrm{PS}}\psi_{p,\sigma}^*(\vec{r}\,')\psi_{p,\rho}(\vec{r}) + \sum_{p\in\mathrm{NS}}\psi_{p,\sigma}^*(\vec{r}\,')\psi_{p,\rho}(\vec{r}), \tag{167}$$

the vacuum-polarization one-particle density matrix in Eq. (159) can be expressed in terms of the vacuum-polarization one-particle density matrix introduced in Eq. (31) as

$$\tilde{n}_{1,\rho\sigma}^{\mathrm{c,vp}}(\vec{r},\vec{r}\,') = \tilde{n}_{1,\rho\sigma}^{\mathrm{vp}}(\vec{r},\vec{r}\,') + n_{1,\rho\sigma}^{\mathrm{c,vp}}(\vec{r},\vec{r}\,'), \tag{168}$$

where we have introduced

$$n_{1,\rho\sigma}^{\mathrm{c,vp}}(\vec{r},\vec{r}\,') = \frac{1}{2}\left(\sum_{p\in\mathrm{NS}}\psi_{p,\sigma}^*(\vec{r}\,')\psi_{p,\rho}(\vec{r}) - \sum_{p\in\mathrm{PS}}\psi_{p,\sigma}^*(\vec{r}\,')\psi_{p,\rho}(\vec{r})\right), \tag{169}$$

which is the vacuum-polarization one-particle density matrix associated with the operator in Eq. (154) but over the free vacuum state, i.e. $\mathbf{n}_1^{\mathrm{c,vp}}(\vec{r},\vec{r}\,') = \langle 0|\hat{\mathbf{n}}_1^{\mathrm{c}}(\vec{r},\vec{r}\,')|0\rangle$. Using charge-conjugation symmetry on the set of eigenfunctions $\{\boldsymbol{\psi}_p\}$ of the free Dirac equation, we have

$$n_{1,\rho\sigma}^{\mathrm{c,vp}}(\vec{r},\vec{r}\,') = \frac{1}{2}\left(\sum_{p\in\mathrm{NS}}\psi_{p,\sigma}^*(\vec{r}\,')\psi_{p,\rho}(\vec{r}) - \sum_{p\in\mathrm{NS}}\sum_{\rho'\sigma'}C_{\rho\rho'}\psi_{p,\sigma'}(\vec{r}\,')\psi_{p,\rho'}^*(\vec{r})C_{\sigma'\sigma}^\dagger\right), \tag{170}$$

or, in matrix form,

$$\mathbf{n}_1^{\text{c,vp}}(\vec{r}, \vec{r}\,') \;\; = \;\; \mathbf{n}_{1,-}^{\text{c,vp}}(\vec{r}, \vec{r}\,') - \mathbf{C}\mathbf{n}_{1,-}^{\text{c,vp}\,\text{T}}(\vec{r}\,', \vec{r})\mathbf{C}^{\dagger}, \tag{171}$$

where $n_{1,-,\rho\sigma}^{\text{c,vp}}(\vec{r}, \vec{r}\,') = (1/2)\sum_{p\in\text{NS}} \psi_{p,\sigma}^{*}(\vec{r}\,')\psi_{p,\rho}(\vec{r})$. We then immediately see that the density associated with $\mathbf{n}_1^{\text{c,vp}}(\vec{r}, \vec{r}\,')$ vanishes

$$n^{\text{c,vp}}(\vec{r}) \;\; = \;\; \text{Tr}[\mathbf{n}_1^{\text{c,vp}}(\vec{r}, \vec{r})] = 0, \tag{172}$$

i.e., the free electron vacuum density and the free positron vacuum density are identical, as already known [25, 56]. Now, using $\mathbf{C}^{\dagger}\boldsymbol{\alpha}\mathbf{C} = \boldsymbol{\alpha}^{\text{T}}$, it can be checked that

$$\sum_{\tau\upsilon} w_{\sigma\tau\rho\upsilon}(\vec{r}_1, \vec{r}_2) n_{\upsilon\tau}^{\text{c,vp}}(\vec{r}_2) = 0, \tag{173}$$

and therefore the contribution of $\mathbf{n}_1^{\text{c,vp}}(\vec{r}, \vec{r}\,')$ to the direct vacuum-polarization potential in Eq. (164) vanishes. Finally, even tough $\hat{\tilde{\mathbf{n}}}_1(\vec{r}_1, \vec{r}_2)$ does not satisfy charge-conjugation symmetry in the sense of Eq. (142), it does satisfy the following relation

$$\hat{\tilde{\mathbf{n}}}_1(\vec{r}_1, \vec{r}_2) = \mathbf{C}\hat{\tilde{\mathbf{n}}}_1^{\text{T}}(\vec{r}_2, \vec{r}_1)\mathbf{C}^{\dagger}, \tag{174}$$

and, together with the symmetry properties of $w_{\sigma\tau\rho\upsilon}(\vec{r}_1, \vec{r}_2)$, it can be used to check that

$$\iint \sum_{\tau\rho\sigma\upsilon} w_{\sigma\tau\rho\upsilon}(\vec{r}_1, \vec{r}_2) n_{1,\upsilon\sigma}^{\text{c,vp}}(\vec{r}_2, \vec{r}_1)\hat{\tilde{n}}_{1,\rho\tau}(\vec{r}_1, \vec{r}_2)\mathrm{d}\vec{r}_1\mathrm{d}\vec{r}_2 = 0, \tag{175}$$

and therefore the contribution of $\mathbf{n}_1^{\text{c,vp}}(\vec{r}, \vec{r}\,')$ to the exchange vacuum-polarization potential in Eq. (165) vanishes as well. This establishes the equivalence between the vacuum-polarization potential in Eq. (38) and in Eq. (163).

The no-particle vacuum energies $\tilde{E}_0$ in Eq. (41) and $\tilde{E}_0^{\text{c}}$ in Eq. (166) are different however. In particular, in comparison to the situation for $\tilde{E}_0$ discussed after Eq. (41), the UV divergences are more serious for $\tilde{E}_0^{\text{c}}$ since the sums in Eq. (166) tend to give cumulative negative energies rather than cancelling energies. For this reason, it might be preferable to work with the electron-positron Hamiltonian $\hat{H}$ in Eq. (7). The form of the electron-positron Hamiltonian $\hat{H}^{\text{c}}$ in Eq. (150) remains useful however to establish links with the literature. In particular, by writing explicitly $\hat{H}^{\text{c}}$ in Eq. (162) in terms of elementary creation and annihilation operators corresponding to the orbital basis $\{\tilde{\psi}_p\}$, and after removing the vacuum energy $\tilde{E}_0^{\text{c}}$, it can be checked that one exactly recovers the effective QED (eQED) Hamiltonian of Refs. [25, 41–45]. So we have

$$\hat{H}_{\text{eQED}} = \hat{H}^{\text{c}} - \tilde{E}_0^{\text{c}} = \hat{H} - \tilde{E}_0, \tag{176}$$

where $\hat{H}_{\text{eQED}}$ is the Hamiltonian in Eq. (46) of Ref. [25]. Whereas this eQED Hamiltonian was obtained in Ref. [25] via a "charge-conjugated contraction" of the fermion operators, here it is obtained via the commutator and anticommutator in Eqs. (154) and (155), or equivalently via the normal ordering with respect to the free vacuum state in Eqs. (11) and (12).

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
