# Peer review of "Relativistic density-functional theory based on effective quantum electrodynamics"

_SciPost Chemistry, doi:SciPost Chem. 1, 002 (2021)_

## Round 1 · Referee Report · Anonymous (Referee 1) · 2021-4-1

Strengths

1 - I very much appreciate that the author in appendices A and B successively demonstrate that the effective QED Hamiltonian (7) possess the correct charge-conjugation symmetry and is equivalent to an alternative Hamiltonian that has been promoted in recent literature.
2 - Focus of the paper is on a relativistic formulation of density-functional theory, but foundations for wave function (or more correctly Fock space)-based theory is given as well.
3 - The DFT section provides foundations as well as important relations for relativistic DFT based on the effective QED Hamiltonian (7).
4 - The author points issues that need further study.
5 - The authors provides a rich bibliography and appears to me fair in his appreciation of these.

Weaknesses

Some points need possibly further sharpening:
1 - First of all, this is not relativistic theory, in the sense of being Lorentz covariant, but rather adheres the pragmatic spirit in which relativistic molecular calculations are carried out presently. This could perhaps be stressed further.
2 - A practical realization of the present work will require the development for the practical realization of regularisation/renormalization procedures.
3 - The authors refers to (opposite) charge for several quantities. This could perhaps be discussed briefly.
4-This is a single author paper. The author adheres to the use of "we", following common practice, but on pages 2, 22 and 25 refers to "my knowledge". I advice being consisten.

Report

This is an absolutely outstanding paper, a veritable treasure trove of important results, which should have very significant impact on the domain of relativistic molecular applications.

Requested changes

The paper can be published as it, but the author should consider the polish suggested above.

  • validity: top
  • significance: top
  • originality: top
  • clarity: top
  • formatting: excellent
  • grammar: excellent

Author:  Julien Toulouse  on 2021-04-29  [id 1390]

(in reply to Report 1 on 2021-04-01)
Category:
reply to objection

The author thanks the referee for the report and the suggestions. Here are replies to the referee's suggestions: 1 - The fact that the present theory is not Lorentz invariant has been added in Section 2.2. 2 - The fact that a practical implementation would require the development of regularization/renormalization procedures has been added in the Conclusions. 3 - We use the opposite charge density because it reduces to the familiar density in the non-relativistic limit (and similarly for other quantities). This explanation has been added in Section 2.2. 4- The language inconsistencies have been corrected.

---

## Round 1 · Referee Report · Paola Gori-Giorgi (Referee 2) · 2021-4-22

Strengths

1- The article puts the foundations of relativistic density functional theory (RDFT) on much firmer grounds than what we had so far, tackling all the details and confusing/misleading arguments of the literature.
2- New exact conditions for the RDFT exchange-correlation functionals are reported.
3- The article is written in a very clear and systematic manner.
4- The article provides clear advancement with respect to state of the art theory on the topic.

Weaknesses

1- A few points are passed over a bit too quick (see suggestions below)

Report

This is a truly excellent article, one of the best I read in the last years. The author put the foundations of relativistic density functional theory (RDFT) on much firmer grounds than what we had so far, tackling all the details and confusing/misleading arguments of the literature. Although no formal proofs are presented, everything is very clear and mostly relies on the same arguments and assumptions used for non relativistic DFT.
All the definitions are very transparent. The author also derives exact properties of the exchange and correlation functionals.
I recommend publication essentially in the present form.
I have only a few small suggestions (reported below).

Requested changes

1- a little discussion on the external potential, i.e., on why it is general enough to assume it to be always of the form of eq. (10);
2- after eq. (61) it would be nice to discuss a bit more about expected features of the exact Hxc potential;
3- eq. (131) is given without much discussion: can something be said about its functional derivative? Is the presence of the absolute value going to create any problem there, why/why not?
4- There is also a typo in line 617: 'examine' -> 'examined'

  • validity: top
  • significance: top
  • originality: good
  • clarity: top
  • formatting: perfect
  • grammar: good

Author:  Julien Toulouse  on 2021-04-29  [id 1391]

(in reply to Report 2 by Paola Gori-Giorgi on 2021-04-22)
Category:
reply to objection

The author thanks the referee for the report and the suggestions. Here are replies to the referee's suggestions:

1- We consider only an external scalar potential and not an external vector potential mostly for simplicity and because this is the most common framework used for molecular calculations. This explanation has been added in Section 2.2.

2- After Eq. (63), we have added the important fact that the Hartree-exchange-correlation (Hxc) potential is both local in space and diagonal in terms of spinor indices (which is unlike in Hartree-Fock theory). Then, we have added at the end of Section 3.2 the fact that each contribution to the potential contains a vacuum-polarization potential term but which are not identical to the vacuum-polarization potentials introduced after Eqs. (39) and (40). The study of the detailed properties of each contribution to the potential is left for future work.

3- The absolute value of the charge density has been used in order to deal with the possibility of having negative charge densities at some points of space in the inhomogeneous system [see discussion in the paragraph after Eq.(55)]. On the one hand, this permits to satisfy charge-conjugation symmetry, but, on the other hand, it introduces discontinuities in the corresponding potential at the points of space where the charge density changes sign. Whether using the absolute value of the charge density is the right thing to do is thus unsure and should be further studied. This discussion has been added in Section 3.5.

4- The typo has been corrected.

---

## Round 2 · Author Response

Dear Editor,

I thank both referees for the reports and suggestions. I have revised the manuscript accordingly.
I hope that the revised manuscript is suitable for publication in SciPost Chemistry.
Sincerely,
Julien Toulouse

---

## Round 2 · List of Changes

• The fact that the present theory is not Lorentz invariant has been added in Section 2.2.

  • The explanation for using the opposite charge density has been added in Section 2.2.

  • This explanation for why we consider only an external scalar potential has been added in Section 2.2.

  • Some information about the Hartree-exchange-correlation potential has been added in Section 3.1 and 3.2.

  • A discussion about the use of the absolute value of the charge density in the LDA functional has been added in Section 3.5.

  • The fact that a practical implementation would require the development of regularization/renormalization procedures has been added in the Conclusions.

  • Some language inconsistencies and typos have been corrected.

  • In addition, a few very minor improvements have been done in various places of the manuscript.

---

## Editorial Decision

published